# Human birth tissue products as a non-opioid medicine to inhibit post-surgical pain

Chi Zhang[1†], Qian Huang[1†], Neil C Ford[1†], Nathachit Limjunyawong[2], Qing Lin[1], Fei Yang[1], Xiang Cui[1], Ankit Uniyal[1], Jing Liu[1], Megha Mahabole[3], Hua He[3], Xuewei Wang[1,4], Irina Duff[1], Yiru Wang[1], Jieru Wan[1], Guangwu Zhu[1], Srinivasa N Raja[1], Hongpeng Jia[5], Dazhi Yang[6], Xinzhong Dong[2,7], Xu Cao[4], Scheffer C Tseng[3], Shaoqiu He[1*], Yun Guan[1,8*]

[1]Department of Anesthesiology and Critical Care Medicine, Johns Hopkins University, School of Medicine, Baltimore, United States; [2]The Solomon H. Snyder Department of Neuroscience, Center for Sensory Biology, Johns Hopkins University, School of Medicine, Baltimore, United States; [3]BioTissue, Inc, Miami, United States; [4]Department of Orthopaedic Surgery, Johns Hopkins University, Baltimore, United States; [5]Department of Surgery, Johns Hopkins University, School of Medicine, Baltimore, United States; [6]Acrogenic Technologies Inc, Rockville, United States; [7]Howard Hughes Medical Institute, Johns Hopkins University, School of Medicine, Baltimore, United States; [8]Department of Neurological Surgery, Johns Hopkins University, School of Medicine, Baltimore, United States

*For correspondence:
shaoqiuh@hotmail.com (SH);
yguan1@jhmi.edu (YG)

[†]These authors contributed equally to this work

## eLife Assessment

The authors provide **convincing** data that identify a novel, non-opioid biologic from human birth tissue products with anti-nociceptive properties in a preclinical mouse model of surgical pain. This **important** study highlights the potential use of naturally derived biologics from human birth tissues as safe and sustainable pain treatment options that do not possess the adverse side effects associated with opioids and synthetic pharmaceuticals. Whether these results will translate to the clinic remains to be seen, nevertheless, these preclinical findings are promising.

**Abstract** Pain after surgery causes significant suffering. Opioid analgesics cause severe side effects and accidental death. Therefore, there is an urgent need to develop non-opioid therapies for managing post-surgical pain. Local application of Clarix Flo (FLO), a human amniotic membrane (AM) product, attenuated established post-surgical pain hypersensitivity without exhibiting known side effects of opioid use in mice. This effect was achieved through direct inhibition of nociceptive dorsal root ganglion (DRG) neurons via CD44-dependent pathways. We further purified the major matrix component, the heavy chain-hyaluronic acid/pentraxin 3 (HC-HA/PTX3) from human AM that has greater purity and water solubility than FLO. HC-HA/PTX3 replicated FLO-induced neuronal and pain inhibition. Mechanistically, HC-HA/PTX3-induced cytoskeleton rearrangements to inhibit sodium current and high-voltage activated calcium current on nociceptive DRG neurons, suggesting it is a key bioactive component mediating pain relief. Collectively, our findings highlight the potential of naturally derived biologics from human birth tissues as an effective non-opioid treatment for post-surgical pain. Moreover, we unravel the underlying neuronal mechanisms of pain inhibition induced by FLO and HC-HA/PTX3.

**eLife digest** Every year, there are about 300 million major surgeries worldwide. Many people experience pain afterward, with some suffering chronic pain. Painkillers commonly used to manage postsurgical pain can have serious side effects. For example, opioid medications, a mainstay therapy for postsurgical pain, can lead to addiction and accidental overdose deaths. Therefore, non-opioid alternatives to manage postsurgical pain are desperately needed.

Substances in human birth tissues may offer a new approach to treating postsurgical pain. Tissue from the umbilical cord and the amniotic membrane are routinely discarded after birth. However, some studies have shown that this tissue can be repurposed as a treatment to promote healing and reduce inflammation in the eye and soft tissues. Studies have also suggested that this tissue may provide pain relief.

Zhang, Huang, Ford et al. showed that tissue derived from discarded post-birth tissues could reduce postsurgical pain in an animal model. In the experiments, mice with an incision on their paw were treated with the birth tissues. The treatment reduced pain in the animals without common side effects seen with opioid treatment. The experiments also provided insights into how the tissue helps relieve pain. The tissue inhibits nerve cells that sense pain through a pathway that requires a protein in the cell membrane called CD44. A compound from the amniotic tissue, called HC-HA/PTX3, is the key ingredient to this pain relief and can, on its own, relieve pain. The compound rearranges the protein scaffolding and reduces sodium and calcium ion signals in the nerve cells, making the cells less active. The experiments suggest a new approach to postsurgical pain control. Because these natural compounds from post-birth tissues have multiple effects, they may help provide improved pain control and help reduce postsurgical hyperactivity of nerve cells that can lead to chronic pain. More studies are needed to confirm the therapy would be safe and effective in humans.

## Introduction

Surgery or trauma may lead to persistent pain, impeding functional recovery and causing considerable distress (**Kehlet et al., 2006**). Continuous reliance on opioid analgesics causes severe side effects and accidental death, which resulted in a national public health emergency being declared in 2017 (**Colvin et al., 2019**). Accordingly, there is an urgent need to develop non-opioid alternative therapies for managing post-surgical/post-operative pain. An optimal strategy would be to develop local treatments that both inhibit pain and address the underlying pathophysiology, such as neuronal sensitization while avoiding the central side effects of commonly used analgesics (**Patapoutian et al., 2009**).

A naturally occurring biologic derived from human birth tissues has recently gained our attention as a potential solution for this challenging problem. The birth tissue is predominantly comprised of the amniotic membrane (AM) and umbilical cord (UC), which share the same cell origin as the fetus. These versatile biological tissues have been used as medical therapy in a wide range of conditions (**Chao et al., 1940**; **de Rotth, 1940**). FLO (Clarix Flo; BioTissue, Miami, FL) is a sterile, micronized, and lyophilized form of human AM/UC matrix used for surgical and non-surgical repair, reconstruction, or replacement of soft tissue by filling in the connective tissue void. They have been shown to orchestrate regenerative healing within its anti-inflammatory and anti-scarring properties in ophthalmic applications (**Tighe et al., 2020**). Intriguingly, FLO also appears to relieve pain effectively in several ocular surface disorders (**Espana et al., 2003**; **Finger, 2008**; **Morkin and Hamrah, 2018**), and musculoskeletal disorders such as osteoarthritis (**Castellanos and Tighe, 2019**; **Mead and Mead, 2020**) and lower extremity neuropathy (**Buksh, 2020**). However, the mechanisms underlying its potential pain-inhibitory properties and how it may affect sensory neuron excitability remain unknown.

In a plantar-incision mouse model of post-surgical pain, we first explored whether FLO may be deployed as a viable biologic for the treatment of trauma pain. We then purified the heavy chain-hyaluronic acid/pentraxin 3 (HC-HA/PTX3), which is in uniquely high amounts in human AM. The natural process of HC-HA/PTX3 formation may involve: Tumor necrosis factor-stimulated gene 6 protein covalently binds to HC1 of inter-alpha-trypsin inhibitor and then transfers it to high-molecular-weight hyaluronan (HMW-HA). At this point, HC1 becomes conjugated, and tumor necrosis factor-stimulated gene 6 is released. PTX3 then tightly associates with the HC1–HA complex by binding to the HC1. HC-HA/PTX3 was recently shown to alleviate dry eye disease induced by chronic graft-versus-host

disease by suppressing inflammation and scarring in murine lacrimal glands (*Ogawa et al., 2017*). We hence investigated whether HC-HA/PTX3 is a key bioactive component mediating the pain relief effects of FLO and further examined its mode of action.

Our findings highlight the potential of naturally derived biologics from human birth tissues as an effective non-opioid treatment for post-surgical pain. Moreover, we unravel the underlying neuronal mechanisms of pain inhibition induced by FLO and HC-HA/PTX3.

## Results

### FLO inhibited post-surgical pain and neuron activation

Intra-paw injection of FLO, but not the vehicle (saline) acutely inhibited heat nociception in naive wild-type (WT) mice (*Figure 1A*). Moreover, FLO dose-dependently (0.1–0.5 mg) attenuated heat hypersensitivity in the hindpaw receiving the plantar-incision (*Figure 1B*), and inhibited mechanical hyperalgesia in the Randall–Sellito test (*Figure 1C*). In the Catwalk assay, FLO partially normalized the impaired gaiting caused by incision, as indicated by increases in print area and max contact area from pre-drug levels (*Figure 1D, E*), suggesting an attenuation of movement-evoked pain which is common after surgery. FLO caused no impairment in locomotor function or exploratory activity in the open-field test (*Figure 1F*); these symptoms are known side effects of opioid use. The concentration of FLO we used fell within the range reported in previous clinical studies (*Ackley et al., 2019*; *Bennett, 2019*; *Castellanos and Tighe, 2019*).

Since nociceptive neuron hyperexcitability may lead to persistent pain (*Xie et al., 2005*; *Zimmermann, 2001*), cellular mechanisms of pain inhibition by FLO can be partly inferred from its modulatory effects on these neurons' excitability. We employed $Pirt^{Cre}$;$Rosa26^{lsl-GCaMP6s}$ mice that exclusively express GCaMP6 (a fluorescent calcium indicator) (*Chen et al., 2013*) in primary sensory neurons to enable high throughput in vivo calcium imaging of dorsal root ganglion (DRG) neurons (*Kim et al., 2008*; *Zheng et al., 2022*; *Figure 2A*). Intra-paw injection of FLO selectively reduced the activation of small-diameter DRG neurons, which are mostly nociceptive neurons, to noxious heat stimulation after plantar-incision (*Figure 2B–D*).

### HC-HA/PTX3 mirrored in vivo pain inhibition by FLO

We then purified HC-HA/PTX3 from the water-soluble extract of human AM (*Figure 3—figure supplement 1A, B*), and biochemically and functionally characterized it using western blot and TRAP assays (*Figure 3—figure supplement 1C–E*; *He et al., 2009*). HC-HA/PTX3 was suggested to be a biologically active component with uniquely high amounts in human birth tissues (*Chen et al., 2015*; *He et al., 2009*; *He et al., 2014*). However, it remains unclear whether HC-HA/PTX3, as a key component, is responsible for FLO's inhibitory effects on neuron activation and pain.

Like FLO, intra-paw injection of HC-HA/PTX3 dose-dependently (10–20 µg) induced heat antinociception in naive WT mice and attenuated heat hypersensitivity developed in the hindpaw after plantar-incision (*Figure 3A*). Moreover, HC-HA/PTX3-induced pain inhibition was comparable between male and female mice after plantar-incision (*Figure 3—figure supplement 2*). HMW-HA, a major component of HC-HA/PTX3, was suggested to attenuate inflammatory pain and neuropathic pain induced by the cancer chemotherapy paclitaxel (*Bonet et al., 2020*; *Bonet et al., 2021b*; *Bonet et al., 2022*; *Ferrari et al., 2018*). We hence examined the effect of HMW-HA on post-surgical pain, which is unknown. We compared the efficacy of HC-HA/PTX3 and HMW-HA, both at an amount of 20 µg since the weight of HC-HA/PTX3 was determined based on its HMW-HA content. At its peak (1 hr post-drug), HC-HA/PTX3 demonstrated a significantly more potent anti-hyperalgesic effect than HMW-HA, and this effect persisted for over 4 hr (*Figure 3B*). In contrast, the effect of HMW-HA had largely dissipated by approximately the 4 hr mark. Additionally, we tested the treatment using a mixture of HMW-HA and HC1, another important component of HC-HA/PTX3. Based on our previous findings, which demonstrated that 1 µg of HC-HA/PTX3 contained 1 µg of HMW-HA, 36 ng of HC1, and 10 ng of PTX3 (*He et al., 2013*), we combined 20 µg of HMW-HA with 720 ng of HC1. However, this mixture did not demonstrate a greater pain-inhibitory effect compared to HMW-HA alone (*Figure 3B*). These findings suggest that the full HC-HA/PTX3 complex is superior to HMW-HA in attenuating post-surgical pain.

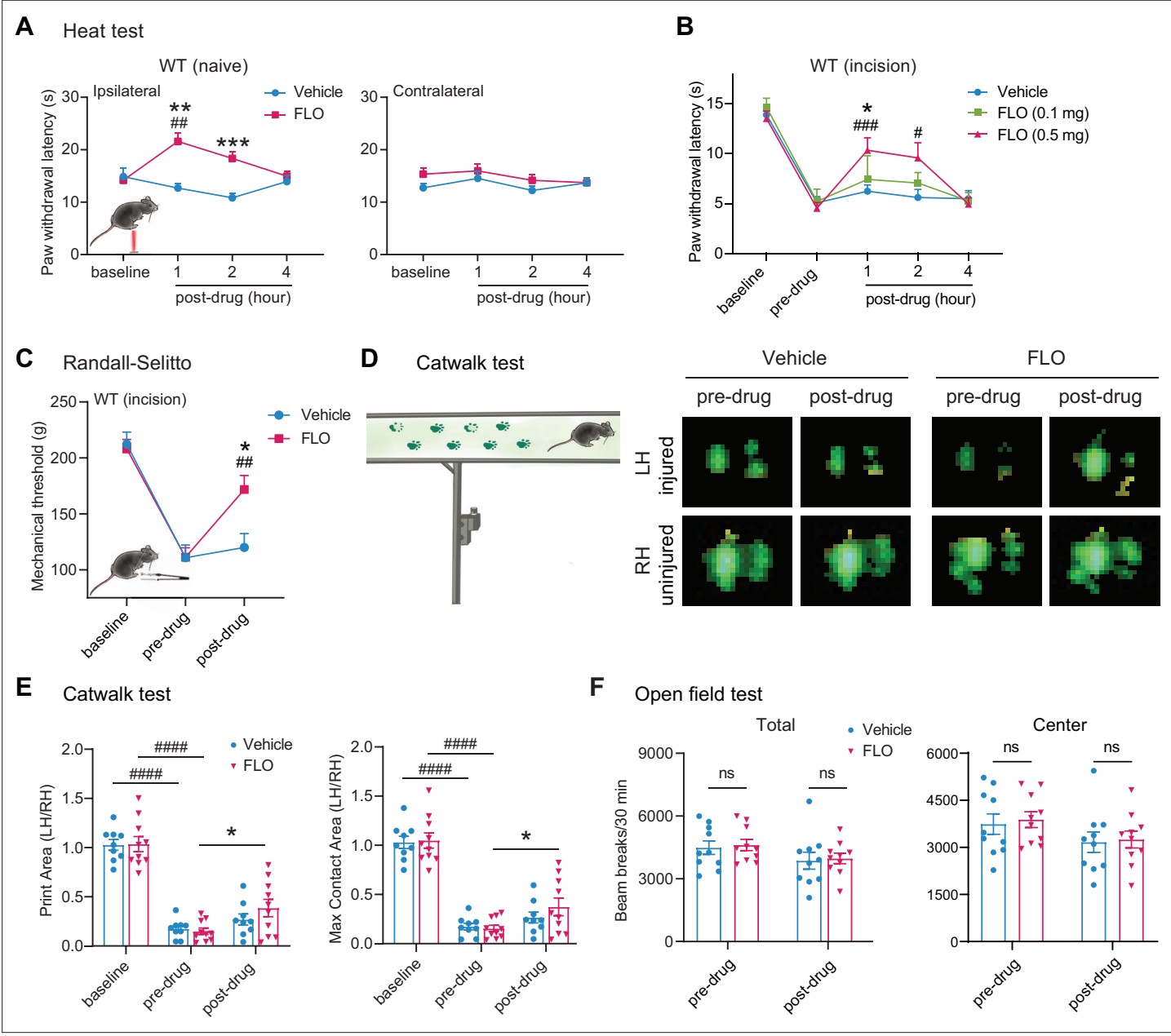

**Figure 1.** Intra-paw injections of FLO inhibited heat nociception in naive wild-type (WT) mice and attenuated both heat and mechanical hyperalgesia after the plantar-incision. (**A**) Paw withdrawal latency (PWL) to heat stimulation in naive WT mice before and after injection of FLO (0.5 mg, 20 μl) or the vehicle (saline, 20 μl) into the dorsum of the hind paw. Ipsilateral: injected side; Contralateral: un-injected side. $N = 10$/group. (**B**) The PWL ipsilateral to the side of the plantar-incision was measured before and 1, 2, and 4 hr after intra-paw injections of FLO (0.1 mg, 0.5 mg, 20 μl) or the vehicle in WT mice during Days 2–4 post-injury. $N = 7$–13/group. (**C**) The mechanical PWT to noxious pinch applied to the side of plantar-incision was measured before and 1 hr after an intra-paw injection of FLO (0.5 mg, 20 μl) or vehicle with the Randall–Selitto test during Days 2–4 post-injury. $N = 10$–11/group. (**D**) Schematic of the Catwalk gait analysis (left) and the representative paw print images (right). (**E**) Quantification of print area and maximum contact area in Catwalk test before and 1 hr after an intra-paw injection of FLO (0.5 mg, 20 μl) or vehicle on Day 2 post-injury. The left hind paw (LH) received the incision and drug treatment, and data were normalized to the right side (RH). $N = 9$–10/group. (**F**) Locomotor function and exploration were assessed in the open field test (30 min duration). The number of total, central, and peripheral beam breaks were measured before and at 1 hr after an intra-paw injection of FLO (0.5 mg, 20 μl) or vehicle during Days 2–4 post-injury. $N = 10$/group. Data are mean ± SEM. Two-way mixed model analysis of variance (ANOVA) followed by Bonferroni post hoc test. (**A–C**) *$p < 0.05$, **$p < 0.01$, ***$p <0.001$ versus vehicle; #$p < 0.05$, ##$p < 0.01$, ###$p < 0.001$ versus baseline (**A**) or pre-drug (**B, C**). (**E, F**) *$p < 0.05$, **$p < 0.01$ versus pre-drug; ###$p < 0.001$, ####$p < 0.0001$ versus baseline. ns = not significant.

The online version of this article includes the following source data for figure 1:

**Source data 1.** Numerical source data files for *Figure 1*.

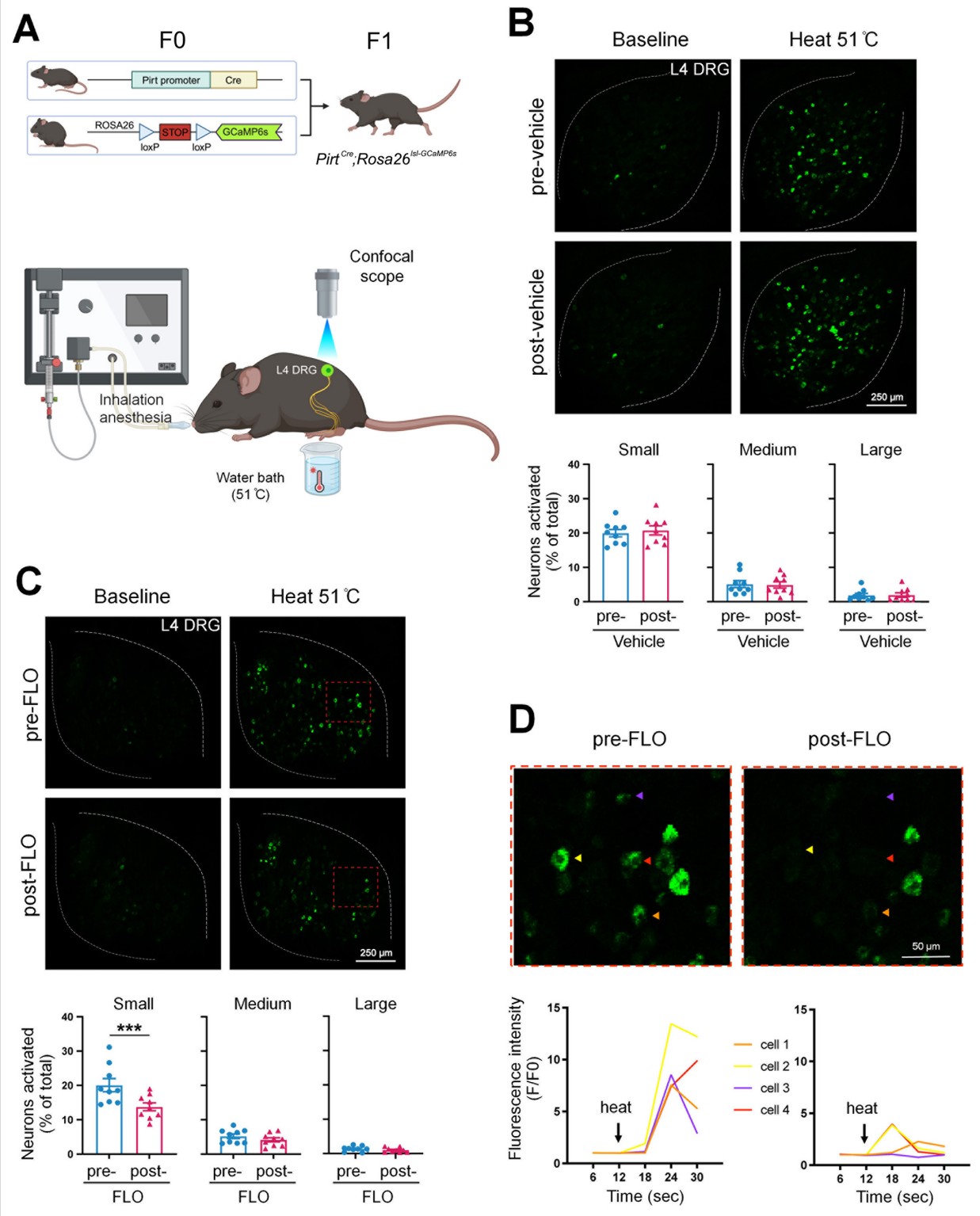

**Figure 2.** FLO acutely attenuated the responses of small dorsal root ganglion (DRG) neurons to noxious heat stimulation. (**A**) Upper: Strategy for generating *Pirt$^{Cre}$;Rosa26$^{lsl-GCaMP6s}$* mice. Lower: The schematic diagram illustrates the experimental setup for in vivo optical imaging of L4 DRG neurons and applying test stimulation. (**B, C**) Upper: Representative images of calcium transients in DRG neurons in response to noxious heat stimulation (51°C water bath) applied to the hind paw before and 1 hr after an intra-paw injection of vehicle (B, saline) or FLO (C, 0.5 mg, 20 µl) at Day 2 after plantar-incision. Lower: Percentages of small-, medium-, and large-size neurons that were activated ($\Delta F/F \geq 30\%$) by heat stimulation before and after vehicle or FLO. '% of total' represented the proportion of activated neurons relative to the total number of neurons counted from the same analyzed image.

*Figure 2 continued on next page*

*Figure 2 continued*

DRG neurons were categorized according to somal size as <450 μm$^2$ (small), 450–700 μm$^2$ (medium), and >700 μm$^2$ (large). *N* = 9/group. (**D**) The higher-magnification representative images (upper) and calcium transient traces (lower) show increased fluorescence intensities in four DRG neurons (indicated by colored arrows) responding to heat stimulation, and decreased responses after FLO treatment. Data are mean ± SEM. (**B, C**) Paired *t*-test. \*\*\*p < 0.001 versus pre-drug.

The online version of this article includes the following source data for figure 2:

**Source data 1.** Numerical source data files for *Figure 2*.

## HC-HA/PTX3 inhibited DRG neurons

Using in vivo calcium imaging of DRG, we have demonstrated that intra-paw injection of FLO significantly reduced the activation of nociceptive neurons in the ganglia to peripheral noxious stimulation (*Figure 2*), possibly through inhibiting the transduction of 'pain' signals at the peripheral nerve terminals, leading to the alleviation of post-surgical pain (*Figure 1*). Due to the technical challenges and limitations of in vivo optical imaging and recording at nerve terminals in skin, we subsequently used cultured DRG neurons to examine the receptor mechanisms and signaling pathways involved in neuron inhibition by FLO.

Since HC-HA/PTX3 mimics FLO in its ability to inhibit pain behavior and has higher purity and greater water solubility compared to FLO, it is well-suited for investigating cellular mechanisms. In cultured lumbar DRG neurons from WT mice, HC-HA/PTX3 exerted inhibitory effects in both calcium imaging (*Figure 3C–F*) and patch-clamp electrophysiology studies (*Figure 3G, H*). Transient receptor potential vanilloid 1 (TRPV1), transient receptor potential ankyrin 1 (TRPA1), and mas-related G-protein-coupled receptor D (MrgprD)-expressing DRG neurons are critical to heat and mechanical pain signaling (*Cavanaugh et al., 2009*). Applying capsaicin (a TRPV1 agonist), cinnamaldehyde (a TRPA1 agonist), or β-alanine (a MrgprD agonist) to bath solution increased intracellular calcium [Ca$^{2+}$]$_i$ in 41%, 24%, or 17% of DRG neurons (*Domocos et al., 2020*; *Tiwari et al., 2016*). Here, HC-HA/PTX3 (15 μg/ml) significantly reduced the [Ca$^{2+}$]$_i$ increase produced by these proalgesics. However, HMW-HA (15 μg/ml) was not effective at the same concentration (*Figure 3C–F*).

Intrinsic membrane excitability (IME) is a property of a neuron that refers to its general state of excitability, which is reflected in part by its ability to generate action potentials (APs). IME of lumbar DRG neurons from WT mice after plantar-incision was measured 24 hr after dissociation. In small DRG neurons, HC-HA/PTX3 concentration-dependently hyperpolarized the membrane potential (5, 10, and 25 μg/ml, *Figure 3G, I*), and increased the rheobase (*Figure 3H, J*), indicating decreased neuronal excitability. However, HC-HA/PTX3 (10 μg/ml) did not significantly affect the IME of large neurons (*Figure 3—figure supplement 3*). These findings suggest that HC-HA/PTX3 may attenuate the activation of key receptors involved in pain control such as TRPV1, TRPA1, and MrgprD. Additionally, it fundamentally decreases nociceptor excitability by inducing membrane hyperpolarization and altering intrinsic membrane properties.

## Pain inhibition by both FLO and HC-HA/PTX3 was CD44-dependent

CD44 is a multifunctional transmembrane glycoprotein that functions as a cell surface adhesion receptor, regulating essential physiologic and pathologic processes (*Campo et al., 2010*; *Janiszewska et al., 2010*). CD44 is the principal receptor for HA (*Ferrari et al., 2018*). Since HMW-HA is a primary component of the HC-HA/PTX3 complex, we explored whether CD44 is required for the inhibition of post-surgical pain by FLO and HC-HA/PTX3.

CD44 is expressed in both neurons and glial cells (*Dzwonek and Wilczynski, 2015*), but its distribution in the DRG was unclear. In mouse DRG, CD44 immuno-reactivity highly colocalized with Na-K ATPase alpha1, a neuron surface marker, but presented at a much lower level in satellite glial cells (*Figure 4—figure supplement 1A*). The specificity of the CD44 antibody was validated in CD44 knockout (KO) mice (*Figure 4—figure supplement 1B*) and by a previous study (*Protin et al., 1999*). CD44 immuno-reactivity highly colocalized with CGRP and IB4, markers of small peptidergic and non-peptidergic nociceptive DRG neurons (*Figure 4A*). Consistent with this finding, analysis of our recently published single-cell RNA-sequencing (scRNA-seq) dataset of mouse DRG showed high expression levels of CD44 in nociceptive neuronal clusters (*Figure 4B*; *Zhang et al., 2022*), but much lower expression in large neuronal clusters which are Aβ or Aδ low-threshold mechanoreceptors or

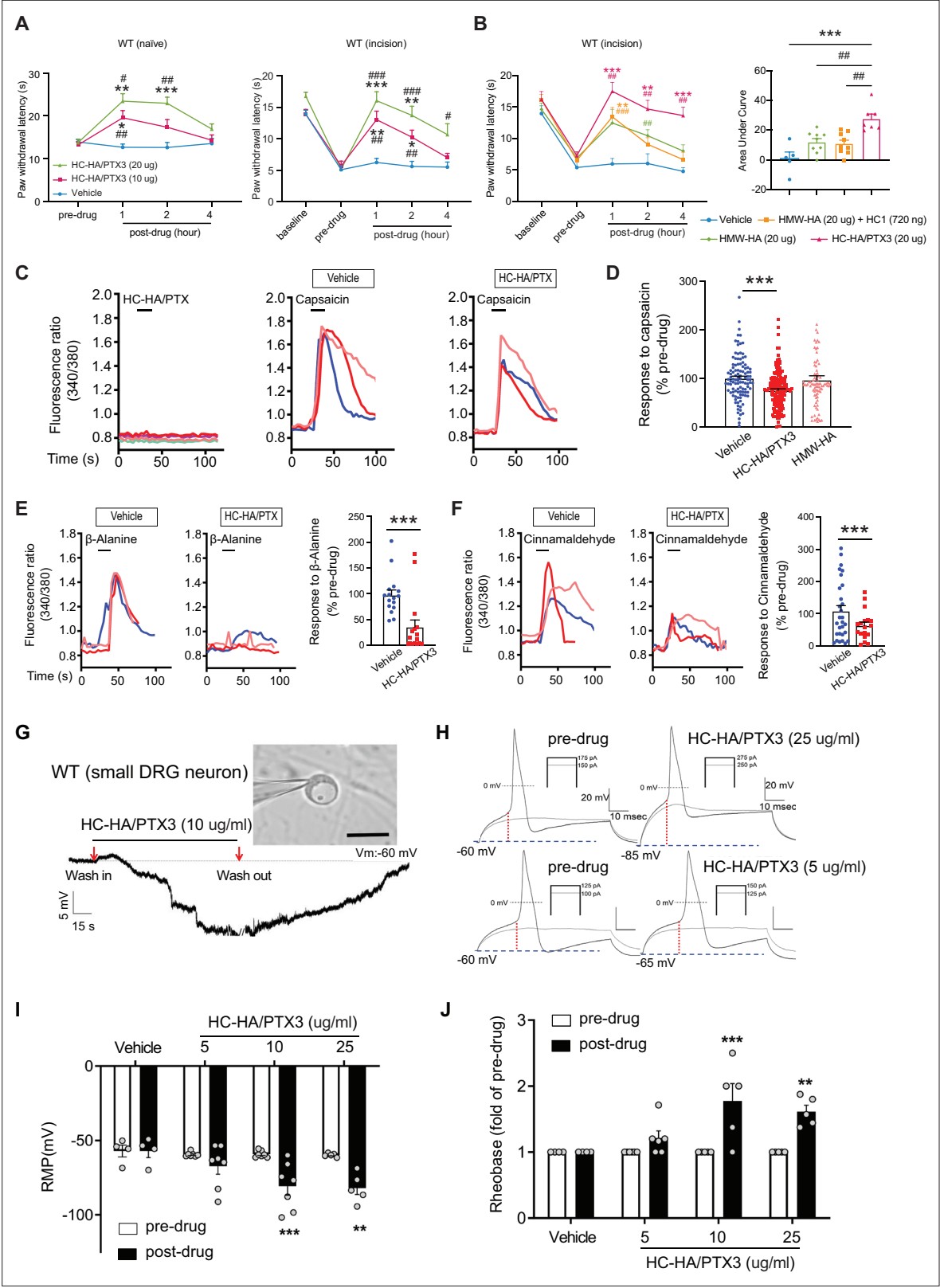

**Figure 3.** Heavy chain-hyaluronic acid/pentraxin 3 (HC-HA/PTX3) inhibited heat hypersensitivity in wild-type (WT) mice after plantar-incision and attenuated dorsal root ganglion (DRG) neuron activation. (**A**) Left: Intra-paw injection of HC-HA/PTX3 (10 μg or 20 μg, 20 μl), but not vehicle (saline), increased paw withdrawal latency (PWL) to heat stimulation in naïve WT mice. *N* = 8–11/group. Right: Intra-paw injection of HC-HA/PTX3 (10 μg or 20 μg, 20 μl) dose-dependently attenuated the heat hypersensitivity during Days 2–4 after plantar-incision. *N* = 9–16/group. (**B**) Right: Intra-paw injection

*Figure 3 continued on next page*

*Figure 3 continued*

of HC-HA/PTX3 (20 µg, 20 µl) showed superior anti-hyperalgesic effect compared to high-molecular-weight hyaluronan (HMW-HA) (20 µg, 20 µl) alone and the mixture of HMW-HA (20 µg) and HC1 (720 ng) during Days 2–4 after plantar-incision. Left: Analyzing the area under the curve (AUC) to assess the anti-hyperalgesic effect of each group. N = 5–9/group. (C) HC-HA/PTX3 inhibited the calcium responses evoked by capsaicin (a TRPV1 agonist, 0.3 µM) in WT DRG neurons. HC-HA/PTX3 alone did not evoke [Ca$^{2+}$]$_i$ elevation. Pretreatment (20 min) of HC-HA/PTX3 (15 µg/ml, bath application) reduced capsaicin-evoked [Ca$^{2+}$]$_i$ rising. (D) The quantification of [Ca$^{2+}$]$_i$ rising evoked by capsaicin in DRG neurons pretreated with the vehicle, HC-HA/PTX3 (15 µg/ml), or HMW-HA (15 µg/ml). N = 109–170 neurons/group. (E) Left: Traces show that the β-alanine (a MrgprD agonist, 1 mM) evoked an increase in [Ca$^{2+}$]$_i$, which was also inhibited by HC-HA/PTX3. Right: The quantification of evoke [Ca$^{2+}$]$_i$ rising by β-alanine. N = 10–25 neurons/group. (F) Left: Traces show that cinnamaldehyde (a TRPA1 agonist, 1 mM) evoked an increase in [Ca$^{2+}$]$_i$, which was inhibited by HC-HA/PTX3. Right: The quantification of evoke [Ca$^{2+}$]$_i$ rising by cinnamaldehyde. N = 15–35 neurons/group. (G) An example trace of membrane potential (Vm) which changed from resting level (−60 mV) toward a more hyperpolarized state after HC-HA/PTX3 (10 µg/ml) in a small DRG neuron (insert, scale bar: 25 µm). Vm returned to pre-drug level after washout. DRG neurons were categorized according to cell body diameter as <20 µm (small), 20–30 µm (medium), and >30 µm (large). (H) Example traces of action potentials (APs) evoked by injection of current in small DRG neurons 5 min after bath application of vehicle or HC-HA/PTX3 (5, 25 µg/ml). (I) HC-HA/PTX3 concentration-dependently altered the intrinsic membrane properties of small DRG neurons. Quantification of the resting membrane potential (RMP) before and at 5 min after bath application of vehicle or HC-HA/PTX3 (5, 10, and 25 µg/ml). N = 4–7/group. (J) Quantification of rheobase in small DRG neurons at 5 min after vehicle or HC-HA/PTX3. The rheobase after the drug was normalized to pre-drug value. N = 5–7/group. Data are mean ± SEM. (A, B: right) Two-way mixed model analysis of variance (ANOVA) followed by Bonferroni post hoc test. *p < 0.05, **p < 0.01, ***p < 0.001 versus vehicle; #p < 0.05, ##p < 0.01, ###p < 0.001 versus pre-drug. (B: left, C) One-way ANOVA followed by Bonferroni post hoc test. ***p < 0.001 versus vehicle; ##p < 0.01 versus other groups. (E, F) Paired *t*-test. ***p < 0.001 versus vehicle. (I, J) Two-way mixed model ANOVA followed by Bonferroni post hoc test. *p < 0.05, **p < 0.01 versus pre-drug.

The online version of this article includes the following source data and figure supplement(s) for figure 3:

**Source data 1.** Numerical source data files for *Figure 3*.

**Figure supplement 1.** Purification and characterization of heavy chain-hyaluronic acid/pentraxin 3 (HC-HA/PTX3).

**Figure supplement 1—source data 1.** PDF file containing original western blots for *Figure 3—figure supplement 1*, indicating the relevant bands and treatments.

**Figure supplement 1—source data 2.** Original files for western blot analysis displayed in *Figure 3—figure supplement 1*.

**Figure supplement 1—source data 3.** Numerical source data files for *Figure 3—figure supplement 1*.

**Figure supplement 2.** Heavy chain-hyaluronic acid/pentraxin 3 (HC-HA/PTX3)-induced comparable inhibition of heat hyperalgesia in male and female mice after plantar-incision.

**Figure supplement 2—source data 1.** Numerical source data files for *Figure 3—figure supplement 2*.

**Figure supplement 3.** Heavy chain-hyaluronic acid/pentraxin 3 (HC-HA/PTX3) did not affect the excitability of large dorsal root ganglion (DRG) neurons in wild-type (WT) mice after the plantar-incision.

**Figure supplement 3—source data 1.** Numerical source data files for *Figure 3—figure supplement 3*.

proprioceptors, and in C-fiber low-threshold mechanoreceptors (c-LTMRs, *Figure 4B*). Another study also demonstrated the high expression of CD44 in nociceptors but not in large DRG neurons (*Vroman et al., 2023*).

Importantly, in cultured lumbar DRG neurons from CD44 KO mice, HC-HA/PTX3 did not reduce the capsaicin-evoked increase of [Ca$^{2+}$]$_i$ (*Figure 4C–E*), nor did it alter the IME and the membrane potential (*Figure 4F, G*). The basal IMEs were comparable between WT and CD44 KO mice (*Supplementary file 1*). Behaviorally, the inhibition of heat hyperalgesia by FLO (0.5 mg, 20 µl) and by HC-HA/PTX3 (10 or 20 µg, 20 µl) was both diminished in CD44 KO mice (*Figure 4H, I*). Likewise, applying a neutralizing antibody (IgG, 10 µg) to CD44, but not control IgG blocked HC-HA/PTX3-induced pain inhibition in WT mice (*Figure 4J*). Collectively, these findings suggest that pain inhibition, as well as the exclusive inhibition of small DRG neurons, which are mostly nociceptive, by HC-HA/PTX3 and FLO, are CD44-dependent.

## HC-HA/PTX3 induced cytoskeletal rearrangement in DRG neurons

We next explored how HC-HA/PTX3 induces CD44-dependent neuronal inhibition. CD44 signaling plays a vital role in regulating the cytoskeleton, forming an intricate fibrous subcellular network that undergoes dynamic changes to regulate cell function (*Föger et al., 2001*; *Freeman et al., 2018*). Immunofluorescence staining showed that HC-HA/PTX3 (10, 15 µg/ml) significantly increased the translocation of F-actin to cell membranes in neurons from lumbar DRG of WT mice (*Figure 5A, B*). However, F-actin fibers were mostly retained in the cytoplasm in HMW-HA-treated (15 µg/ml) group.

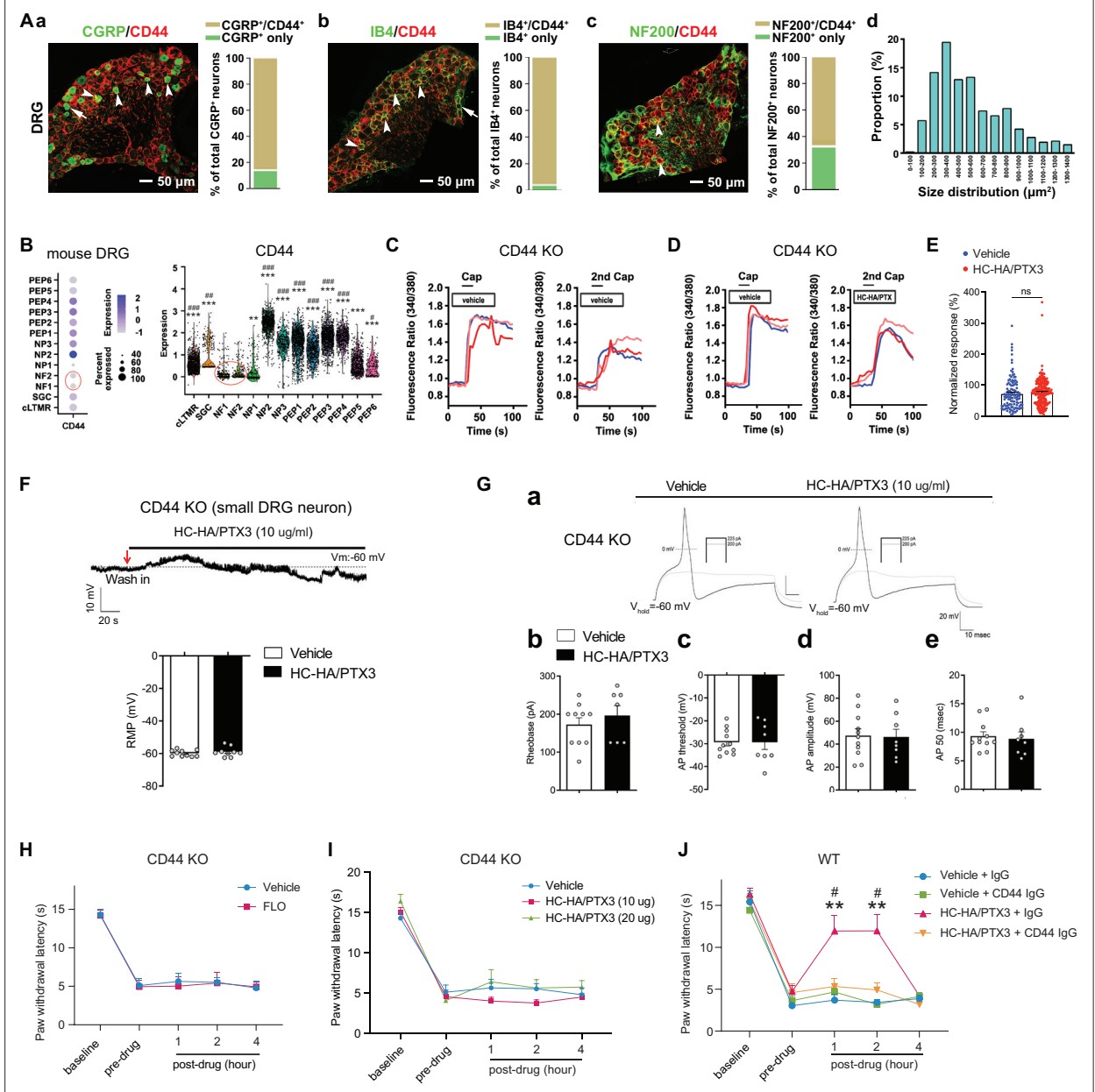

**Figure 4.** FLO and heavy chain-hyaluronic acid/pentraxin 3 (HC-HA/PTX3) inhibited pain via CD44-dependent mechanisms. (**A**) The expression of CD44 in the dorsal root ganglion (DRG) of wild-type (WT) mice. Left: Colocalization of CD44 and CGRP (**a**), IB4 (**b**), and NF200 (**c**) immunoreactivity (IR). Right: The quantification of CD44-expressing neurons (as % of total neurons in each subpopulation, IB4+: 96%; CGRP+: 82%; NF200+: 68%, N = 4). (**d**) The size distribution of CD44-expressing neurons. (**B**) Left: Dot plot of CD44 gene expression in different clusters [SGC (1), NF (2), NP (3), PEP (6), cLTMR (1)] of DRG cells from WT mice in single-cell RNA-sequencing study. The dot size represents the percentage of cells expressing CD44, and the color scale indicates the average normalized expression level. The NF1 and NF2 clusters were indicated with a red circle. Right: Violin plot shows the CD44 expression levels in each cluster. SGC: satellite glial cells; NF, Aβ or Aδ low-threshold mechanoreceptors or proprioceptors; NP, non-peptidergic nociceptors or pruriceptors; PEP, peptidergic nociceptors; C-LTMR, C-fiber low-threshold mechanoreceptors. One-way analysis of variance (ANOVA) followed by Bonferroni post hoc test. **p < 0.01, ***p < 0.001 versus NF1; #p < 0.05, ##p < 0.01, ###p < 0.001 versus NF2. (**C**) Traces show that the capsaicin (0.3 μM) evoked an increase of [Ca2+]i in a small neuron from a CD44 KO mouse. Compared to [Ca2+]i rising evoked by the first capsaicin application, there was a reduction of [Ca2+]i rising to the second treatment, indicating TRPV1 desensitization. DRG neurons were categorized according to cell body diameter as <20 μm (small), 20–30 μm (medium), and >30 μm (large). (**D**) Capsaicin-evoked increases of [Ca2+]i before and after treatment (20 min) with HC-HA/PTX3 (10 μg/ml) in small DRG neurons from CD44 KO mice.(**E**) The quantification of evoked [Ca2+]i rising by capsaicin. HC-HA/PTX3 pretreatment did not reduce capsaicin-evoked [Ca2+]i rising in CD44 KO neurons. N = 100–120 neurons/group. (**F**) HC-HA/PTX3 did not change the intrinsic membrane property of small DRG neurons from CD44 KO mice. Upper: An example trace of membrane potential (Vm) which remained around resting level (−60 mV) after HC-HA/PTX3 (10 μg/ml). Lower: Quantification of the resting membrane potential (RMP) at 5 min after vehicle (saline) and

*Figure 4 continued on next page*

*Figure 4 continued*

HC-HA/PTX3 (p = 0.48). (**G**) Upper: Examples of traces of action potentials (APs) and rheobase evoked by injection of current in a small CD44 KO DRG neuron at 5 min after vehicle or HC-HA/PTX3 (10 μg/ml). Lower: Quantification of the rheobase levels (p = 0.2), AP threshold (p = 0.87), AP amplitude (p = 0.75), and duration (p = 0.82) in small DRG neurons from CD44 KO mice. N = 7–11/group. (**H**) Paw withdrawal latency (PWL) that was ipsilateral to the side of plantar-incision before and after an intra-paw injection of FLO (0.5 mg, 20 μl) or vehicle (saline, 20 μl) in CD44 KO mice (H, N = 8–9/group) after plantar-incision. (**I**) The ipsilateral PWL before and after an intra-paw injection of HC-HA/PTX3 (10 μg or 20 μg, 20 μl) or vehicle in CD44 KO mice after plantar-incision. N = 7–9/group. (**J**) The ipsilateral PWL before and after intra-paw injection of vehicle + control IgG, vehicle + CD44 IgG, HC-HA/PTX3 (10 μg) + control IgG, or HC-HA/PTX3 (10 μg) + CD44 IgG (all IgG at 10 μg, 10 μl) in WT mice after plantar-incision. N = 8–11/group. Data are mean ± SEM. (**E**) One-way ANOVA followed by Bonferroni post hoc test. ns = not significant. (**F, G**) Student's *t*-test. (**H–K**) Two-way mixed model ANOVA followed by Bonferroni post hoc test. **p < 0.01 versus vehicle or saline + IgG; #p < 0.05 versus pre-drug.

The online version of this article includes the following source data and figure supplement(s) for figure 4:

**Source data 1.** Numerical source data files for *Figure 4*.

**Figure supplement 1.** The expression of CD44 in mouse dorsal root ganglion (DRG).

**Figure supplement 1—source data 1.** PDF file containing original western blots for *Figure 4—figure supplement 1*, indicating the relevant bands and treatments.

**Figure supplement 1—source data 2.** Original files for western blot analysis displayed in *Figure 4—figure supplement 1*.

HC-HA/PTX3 at the concentration (0–15 μg/ml) tested did not induce neuronal toxicity, per the MTT (3-(4,5-dimethylthiazol-2-yl)-2,5-diphenyltetrazolium bromide) assay (*Figure 5C*).

Increased sub-membranous F-actin after HC-HA/PTX3 treatment suggests greater availability of cortical actin filaments, accompanied by increased translocation of CD44 to the cell membrane (*Figure 5A, B*). These effects of HC-HA/PTX3 were prevented by a bath application of latrunculin-A (LAT-A, 1 μM, *Figure 5A, B*), an actin polymerization inhibitor that compromises the integrity of the cytoskeleton (*Fujiwara et al., 2018*). Moreover, knocking down of profilin-1 (*Pfn1*), an essential element for promoting actin polymerization (*Alkam et al., 2017*; *Witke, 2004*), by electroporating DRG neurons with siRNA specifically targeting *Pfn1* (si*Pfn1*) also diminished HC-HA/PTX3-induced cytoskeletal rearrangement (*Figure 5D*).

## HC-HA/PTX3-induced cytoskeletal rearrangement depended on CD44 signaling

Notably, pretreatment with CD44 IgG (2 μg/ml) blocked the cytoskeletal rearrangement induced by HC-HA/PTX3 (15 μg/ml, *Figure 5E, F*). In line with this finding, the condensation of sub-membranous F-actin and translocation of CD44 to the cell membrane after HC-HA/PTX3 were significantly increased only in small DRG neurons (*Figure 5—figure supplement 1*), which express much higher levels of CD44 than large neurons (*Figure 4A, B*).

We next explored the downstream intracellular components involved in CD44-dependent cytoskeletal rearrangement induced by HC-HA/PTX3, which remain unknown. CD44 may activate cytoskeletal proteins, such as *Ank2*-encoded Ankyrin-B and *Ank3*-encoded Ankyrin-G, which are highly expressed in DRG neurons and can modulate neuronal excitability (*Stevens and Rasband, 2021*). CD44 can also interact with cortical actin filaments via ezrin/radixin/moesin (ERM) signaling (*Fehon et al., 2010*). Accordingly, we knocked down *Ank2* and *Ank3* together as siAnk group, and *Ezr*, *Rdx*, and *Msn* together as siERM group in cultured WT DRG neurons (*Figure 5—figure supplement 2A, B*). The mRNA levels of targeted genes were significantly decreased, but not abolished, after electroporation with the specific siRNAs (*Figure 5—figure supplement 2C*). Although HC-HA-PTX3 still increased the translocation of F-actin and CD44 to the cell membrane in both siAnk and siERM groups, the extent was significantly less than that in the siNT control group (*Figure 5—figure supplement 2A, B*), suggesting that both Ankyrin and ERM signaling may partly contribute to HC-HA/PTX3-induced cytoskeletal rearrangement.

## Cytoskeletal rearrangement contributed to the inhibition of ion channels by HC-HA/PTX3

F-actin constitutes a sub-membranous cytoskeleton network, serving as an important scaffold for membrane ion channels, receptors, and intracellular kinases to function properly (*Vasilev et al., 2021*). Analgesic mechanisms often involve modulation of ion channels, especially inhibiting high-voltage

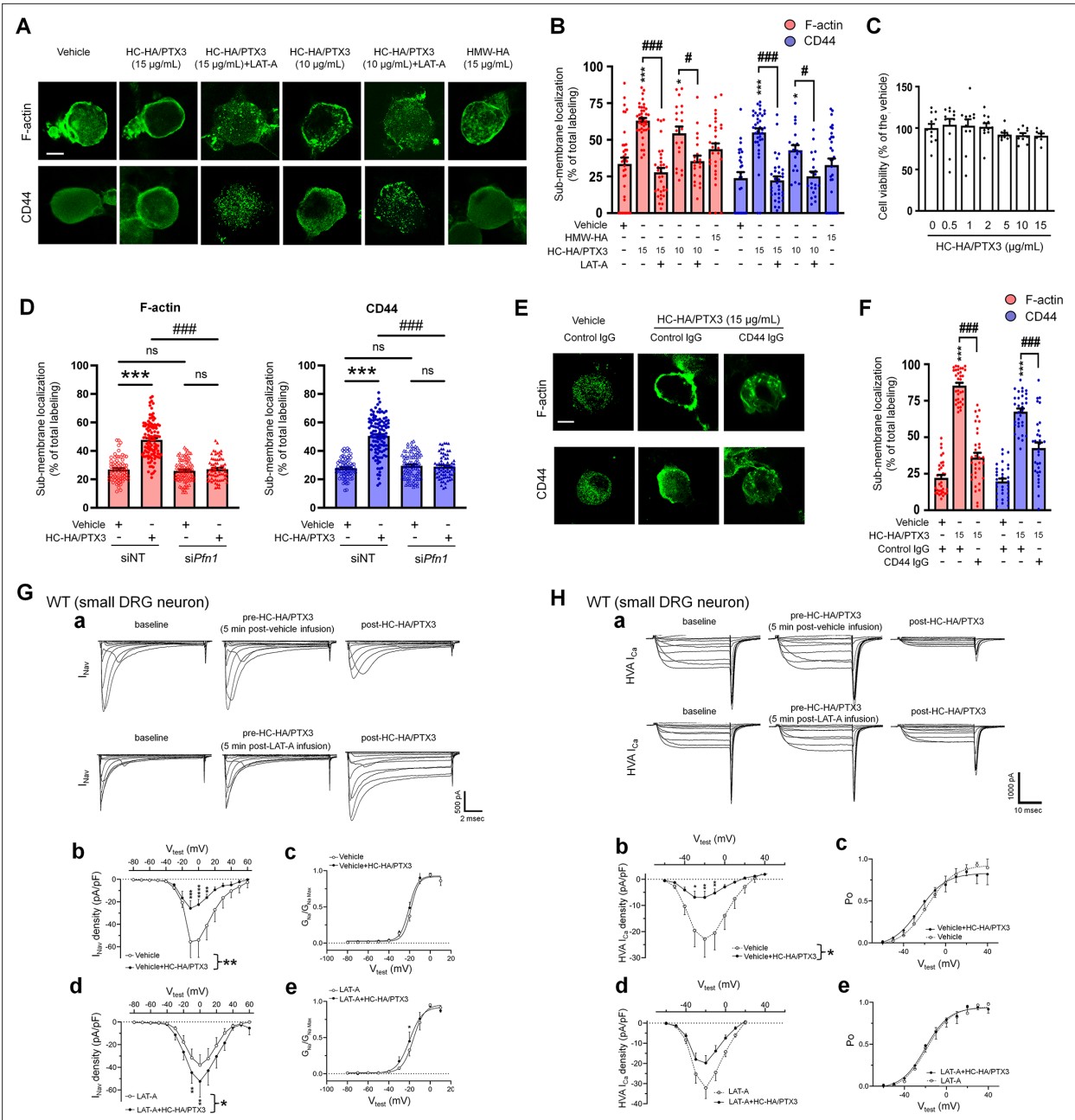

**Figure 5.** Heavy chain-hyaluronic acid/pentraxin 3 (HC-HA/PTX3) induced cytoskeletal rearrangement which contributed to its inhibition of $I_{Nav}$ and HVA-$I_{Ca}$. (**A**) Example images show the distribution of F-actin and CD44 staining in small dorsal root ganglion (DRG) neurons of wild-type (WT) mice. Neurons were treated with bath application of vehicle (saline), high-molecular-weight hyaluronan (HMW-HA) (15 µg/ml), HC-HA/PTX3 (10, 15 µg/ml), or HC-HA/PTX3 (10, 15 µg/ml) combined with Latrunculin A (LAT-A, 1 µM) for 45 min. Scale bar: 5 µm. DRG neurons were categorized according to cell body diameter as <20 µm (small), 20–30 µm (medium), and >30 µm (large). (**B**) Quantification of sub-membranous F-actin polymerization and translocation of CD44 in small WT DRG neurons after drug treatment. $N$ = 30–80/group. (**C**) Proliferation MTT assay showed a lack of neuronal toxicity from 0.5, 1, 2, 5, 10, and 15 µg/ml HC-HA/PTX3, compared to vehicle (100% viable cells). $N$ = 6–12 repetitions/group. (**D**) Quantification of sub-membranous F-actin polymerization and translocation of CD44 in small DRG neurons. DRG neurons were electroporated with siRNA targeting *Pfn1* (si*Pfn1*) or non-targeting siRNA (siNT, control). Neurons were treated with vehicle (saline) or HC-HA/PTX3 (10 µg/ml) for 45 min. $N$ = 70–111/group. (**E**) Changes in the sub-membrane distribution of F-actin and CD44 in WT DRG neurons treated with vehicle + control IgG (2 µg/ml), HC-HA/PTX3 (15 µg/ml) + control IgG (2 µg/ml), or HC-HA/PTX3 (15 µg/ml) + CD44 IgG (2 µg/ml) for 45 min. Scale bar: 5 µm. (**F**) Quantification of the sub-membrane F-actin and CD44 labeling in each group. (**G**) Infusion of LAT-A attenuated the inhibition of $I_{Nav}$ by HC-HA/PTX3 in WT DRG neurons. (a) Representative traces of $I_{Nav}$ after 5 min infusions of vehicle (top row) or LAT-A (bottom row, 0.5 nM) through the recording electrode, followed by bath application of HC-HA/PTX3 (10 µg/ml). Lumbar DRG neurons were harvested on Days 2–3 after plantar-incision. (b) There was a significant interaction between the variation produced by

*Figure 5 continued on next page*

*Figure 5 continued*

HC-HA/PTX3 (10 µg/ml) and test voltages ($V_{Test}$) applied in vehicle-infused neurons, resulting in an overall $I_{Nav}$ inhibition ($F_{(14,90)}$ = 3.29, ***$p$ < 0.001), and significantly decreased $I_{Nav}$ density (pA/pF) from $V_{Test}$ = −10 to +10 mV, as compared to pre-HC-HA/PTX3 treatment. N = 7/group. (c) HC-HA/PTX3 did not alter $G_{Na}/G_{Na}$ max across the test voltages ($F_{(9,60)}$ = 0.44, $p$ = 0.9) in vehicle-infused neurons. N = 7/group. (d) There was a significant interaction between the variation produced by HC-HA/PTX3 (10 µg/ml) and $V_{Test}$ applied in LAT-A-infused neurons, resulting in overall $I_{Nav}$ increase ($F_{(14,120)}$ = 1.87, *$p$ < 0.05) and increased $I_{Nav}$ density (pA/pF) from $V_{Test}$ = −10 to 0 mV, as compared to pre-HC-HA/PTX3. N = 9/group. (e) HC-HA/PTX3 significantly increased the $G_{Na}/G_{Na}$ max at $V_{Test}$ = −20 mV in LAT-A-infused neurons (*$p$ < 0.05, N = 9/group). (H) LAT-A attenuated the inhibition of HVA-$I_{Ca}$ by HC-HA/PTX3 in WT DRG neurons. (a) Representative traces of HVA-$I_{Ca}$ in small WT DRG neurons after 5 min infusions of vehicle (top row) or LAT-A (bottom row, 0.5 nM), followed by bath application of HC-HA/PTX3 (10 µg/ml). (b) In vehicle-infused neurons, HC-HA/PTX3 (10 µg/ml) significantly decreased HVA-$I_{Ca}$ ($F_{(1,12)}$ = 6.52, *$p$ = 0.02) and HVA-$I_{Ca}$ conductance ($I/I_{max}$) from $V_{Test}$ = −40 to +10 mV, as compared to pre-HC-HA/PTX3. N = 7. (c) HC-HA/PTX3 did not alter the channel open probability (Po) in vehicle-infused neurons ($p$ = 0.82, N = 7). (d) In LAT-A-infused neurons, HC-HA/PTX3 only modestly reduced HVA-$I_{Ca}$ conductance across test voltages applied ($F_{(1,12)}$ = 0.27, $p$ = 0.6, N = 8). (e) HC-HA/PTX3 did not alter Po in LAT-A-infused neurons ($p$ = 0.94, N = 8). Data are mean ± SEM. (B–D, F) One-way analysis of variance (ANOVA) followed by Bonferroni post hoc test. *$p$ < 0.05, ***$p$ < 0.001 versus vehicle; #$p$ < 0.05, ###$p$ < 0.001 versus indicated group. (G, H) Two-way repeated measures ANOVA with Holm–Sidak post-test. *$p$ < 0.05, **$p$ < 0.01, ***$p$ < 0.001, ****$p$ < 0.0001 versus vehicle infusion or LAT-A infusion group.

The online version of this article includes the following source data and figure supplement(s) for figure 5:

**Source data 1.** Numerical source data files for *Figure 5*.

**Figure supplement 1.** Differential effects of heavy chain-hyaluronic acid/pentraxin 3 (HC-HA/PTX3) on sub-membranous F-actin polymerization and translocation of CD44 in small, medium, and large wild-type (WT) dorsal root ganglion (DRG) neurons.

**Figure supplement 1—source data 1.** Numerical source data files for *Figure 5—figure supplement 1*.

**Figure supplement 2.** Quantification of sub-membranous F-actin polymerization and translocation of CD44 in small wild-type (WT) dorsal root ganglion (DRG) neurons in each group.

**Figure supplement 2—source data 1.** Numerical source data files for *Figure 5—figure supplement 2*.

**Figure supplement 3.** The inhibitions of $I_{Nav}$ by heavy chain-hyaluronic acid/pentraxin 3 (HC-HA/PTX3) in small dorsal root ganglion (DRG) neurons were diminished in CD44 KO mice and were inhibited by a pretreatment of LAT-A in neurons from WT mice.

**Figure supplement 3—source data 1.** Numerical source data files for *Figure 5—figure supplement 3*.

**Figure supplement 4.** Intracellular infusion of LAT-A did not change the gross morphology of dorsal root ganglion (DRG) neurons in patch-clamp recordings.

activated (HVA) calcium current ($I_{ca}$, e.g., morphine) and sodium current ($I_{Nav}$, e.g., lidocaine) (*Garber, 2005*; *Rogawski and Löscher, 2004*; *Schroeder and McCleskey, 1993*). HC-HA/PTX3 (10 µg/ml) inhibited depolarization-elicited $I_{Nav}$ in small neurons from the lumbar DRG in WT mice at Days 2–4 after plantar-incision, compared to the vehicle (*Figure 5G*). In contrast, the same treatment increased $I_{Nav}$ in that of CD44 KO mice (*Figure 5—figure supplement 3A, B*). Importantly, pretreatment with an intracellular infusion of LAT-A (0.5 nM) abolished the inhibition of $I_{Nav}$ by HC-HA/PTX3 in WT mice; instead, an increased $I_{Nav}$ was observed (*Figure 5G*, *Figure 5—figure supplement 3B*). LAT-A itself minimally affected $I_{Nav}$. HC-HA/PTX3 also inhibited the HVA-$I_{ca}$ current in WT DRG neurons. This effect was also attenuated by intracellular infusion of LAT-A (*Figure 5H*, *Figure 5—figure supplement 3C*). Collectively, these findings suggest that HC-HA/PTX3 may inhibit cell membrane ion channels through a CD44-mediated cytoskeleton rearrangement. Despite the cytoskeletal effects, neither bath application of HC-HA/PTX3 (15 µg/ml) nor intracellular infusion of LAT-A (0.5 nM) changed the gross morphology of DRG neurons (*Figure 5—figure supplement 4*).

## Discussion

Our study revealed that human birth tissue products (FLO) mitigated post-surgical pain by directly inhibiting nociceptive DRG neurons. The major matrix component purified from human AM, HC-HA/PTX3, mimics the pain- and neuronal-inhibitory effects of FLO. Mechanistically, both compounds exert pain inhibition in a CD44-dependent manner. At the cellular level, HC-HA/PTX3 caused membrane hyperpolarization, modified the intrinsic properties, and inhibited $I_{Nav}$ and HVA-$I_{ca}$ in nociceptive DRG neurons through cytoskeleton rearrangement and interaction with these membrane ion channels.

Cryopreserved AM/UC has been clinically validated through numerous studies since 1995, including clinical trials specifically assessing FLO (Clarix Flo). These studies collectively support the safety and preliminary effectiveness of FLO in managing some clinical pain conditions such as knee osteoarthritis (*Castellanos and Tighe, 2019*; *Mead and Mead, 2020*), discogenic pain (*Buck, 2019*),

rotator cuff tears (*Ackley et al., 2019*), and painful neuropathy of the lower extremities (*Buksh, 2020*). These clinical findings also provide important premise of our mechanistic study of FLO and HC-HA/PTX3 in the treatment of post-surgical pain. Applying FLO locally inhibited heat and mechanical hyperalgesia, and attenuated movement-induced pain in WT mice following plantar-incision. Similarly, HC-HA/PTX3 also inhibited heat hyperalgesia. Intriguingly, treatment with HC-HA/PTX3 demonstrated a stronger and longer-lasting pain-inhibitory effect compared to treatment with HMW-HA and the combination of HMW-HA and HC1, at their comparable dosages tested. This finding suggests the superiority of HC-HA/PTX3 over HMW-HA for post-surgical pain treatment, yet the underlying mechanisms remain unclear. After in vivo administration, HMW-HA undergoes rapid and progressive degradation through a series of enzymatic reactions, resulting in the formation of polymers of decreasing sizes and altered bioactivity. Notably, different sizes of HA fragments exhibit a wide range of biological functions, which are sometimes opposing. For example, larger hyaluronan polymers (e.g., HMW-HA) possess anti-angiogenic, immunosuppressive, and anti-hyperalgesic properties. In contrast, smaller polysaccharide fragments are often inflammatory, immuno-stimulatory, angiogenic, and proalgesic. Moreover, they can compete with larger hyaluronan polymers for target receptors (*Stern et al., 2006*). The long-lasting drug action of HC-HA/PTX3 in vivo indicates that it may possess a more stable structure than HMW-HA, making it less prone to degradation and loss of bio-efficacy.

Chronic post-surgical pain, which persists for at least 3 months after surgery, is most difficult to treat. In addition to the plantar-incision model, other chronic post-surgical pain models, such as skin/muscle incision and retraction model and laparotomy model (*Flatters, 2008*; *Martinez et al., 2024*), as well as neuropathic pain models, also need to be tested for treatment with FLO and HC/HA/PTX3 in future studies to extend current findings and improve the translational potential.

Small DRG neurons are activated by heat and noxious stimulation, while innocuous mechanical stimulation mainly activates large DRG neurons and low-threshold mechanoreceptors (LTMRs) (*Qi et al., 2024*). Both FLO and HC-HA/PTX3 inhibited small DRG neurons, as evidenced by decreased excitability in functional calcium imaging and electrophysiologic recording. We found that the potent inhibitory effect of HC-HA/PTX3 on small DRG neurons may result from their high expression of CD44. Both pain- and neuronal-inhibitory effects from FLO and HC-HA/PTX3 were CD44-dependent. Importantly, scRNA-seq analysis showed a significantly higher expression of CD44 in small nociceptive DRG neurons, compared to large neurons (e.g., LTMRs and proprioceptors). Similar findings were also observed in DRG scRNA-seq datasets from another two studies (*Renthal et al., 2020*; *Vroman et al., 2023*). In line with this notion, HC-HA/PTX3 exerted a minimal effect on large DRG neurons. Nevertheless, the mechanisms for the differential drug effects warrant further investigations.

Unlike traditional analgesics, which often target a single downstream effector, HC-HA/PTX3 may induce a range of changes that could fundamentally decrease nociceptor excitability. These changes include membrane hyperpolarization, altered intrinsic membrane properties, and the inhibitions of multiple membrane ion channels, including $I_{Nav}$ and HVA-$I_{ca}$. In addition, HC-HA/PTX3 reduced calcium responses evoked by several proalgesic compounds, including capsaicin, β-alanine, and cinnamal in nociceptive DRG neurons. The activations of TRPV1, MrgprD, and TPRA1 are known to be important to heat and mechanical hypersensitivity and themselves are important targets for pain control. Therefore, HC-HA/PTX3 may effectively block the transmission of noxious afferent inputs through multiple modes of action.

To explore the molecular basis of HC-HA/PTX3-induced neuronal inhibition, our study unraveled that HC-HA/PTX3, but not HMW-HA, induced a rearrangement of the cytoskeleton, leading to an increase in sub-membranous F-actin polymerization and the translocation of CD44 to the vicinity of the cell membrane in small DRG neurons. This effect was not observed in large DRG neurons, and was blocked by both LAT-A pretreatment and *Pfn1* knockdown, which disrupt actin polymerization. Given that polymerized F-actin serves as a scaffold for signaling that affects ion channels, we speculate that the cytoskeleton rearrangement triggered by HC-HA/PTX3 could thus profoundly change ion channel function and, consequently, neuronal excitability. Supporting this hypothesis, the intracellular infusion of a low dose of LAT-A, which had minimal effect on $I_{Nav}$, and HVA-$I_{ca}$ and the morphology of DRG neurons, blocked the inhibition of $I_{Nav}$ and HVA-$I_{ca}$ by HC-HA/PTX3. Thus, HC-HA/PTX3 may inhibit ion channels through a CD44-mediated cytoskeleton rearrangement, representing a novel mechanism for neuron inhibition.

The signaling of HMW-HA also depends on CD44 clustering in lipid rafts, and disrupting this markedly reduces HMW-HA-induced anti-hyperalgesia (*Bonet et al., 2020*). However, at the concentration tested (15 µg/ml), HMW-HA did not increase cortical F-actin and CD44 translocation in small DRG neurons, nor attenuated capsaicin-induced $[Ca^{2+}]_i$ increase. These findings suggest that different ligands may induce varying cellular effects after binding to CD44. Indeed, HA of different molecular weights can activate different downstream signaling pathways of CD44, leading to opposing effects. For example, HMW-HA produced anti-hyperalgesia, while low-molecular-weight HA- (LMW-HA) induced hyperalgesia (*Ferrari et al., 2018*). While Src signaling was involved in LMW-HA-induced hyperalgeisa (*Bonet et al., 2020*), other downstream signaling pathways of CD44s may participate the anti-hyperalgesic effect of HMW-HA, including phosphatidylinositol (PI) 3-kinase gamma (PI3Kγ)/ protein kinase B (AKT), RhoGTPases (RhoA and Rac1), and phospholipases (phospholipases Cε and Cγ1) (*Bonet et al., 2020*; *Bonet et al., 2021b*; *Bonet et al., 2022*). These findings suggest a complex interplay of downstream signaling pathways of CD44 in neuronal and pain modulation.

HMW-HA was reported to attenuate CIPN only in male rats (*Bonet et al., 2022*). However, sex dimorphism was not observed in the inhibition of PGE2-induced inflammatory pain by HMW-HA (*Bonet et al., 2020*). Similarly, HC-HA/PTX3-induced comparable pain inhibition in both sexes. For LMW-HA-induced hyperalgesia, three receptors, including CD44, toll-like receptor 4 (TLR4), and receptor for hyaluronan-mediated motility (RHAMM), may be involved. Yet, estrogen dependence was only established for RHAMM-dependent hyperalgesia and its inhibition by HMW-HA (*Bonet et al., 2021a*) These findings collectively suggest a complex interplay between estrogen and different types of HA and HA receptors in pain regulation.

## Conclusions

Our study suggests that human birth tissue products may be deployed as a viable biologic to treat post-surgical pain. We further identified HC-HA/PTX3 as the primary bioactive component responsible for pain inhibition. It induced an acute cytoskeleton rearrangement and inhibition of $I_{Nav}$ and HVA-$I_{ca}$ currents in a CD44-dependent manner, making it a promising non-opioid treatment for post-surgical pain. Nevertheless, translation of preclinical findings into clinical treatments is complex and often challenging due to species differences and the intricate nature of pain etiology in patients. Our findings provide an important rational for future clinical trials to validate the utility of FLO and HC-HA/PTX3 for post-surgical pain control.

# Materials and methods

## Animals

C57BL/6 mice and CD44 KO mice (B6.129(Cg)-Cd44tm1Hbg/J, strain #005085) (*Protin et al., 1999*) were purchased from Jackson (Jax) Laboratories. The *Pirt^Cre* mice and *Rosa26^lox-stop-lox-GCaMP6s* (*Rosa26^lsl-GCaMP6s*) mice were generated by Dr. Xinzhong Dong in the Solomon H Snyder Department of Neuroscience, School of Medicine, Johns Hopkins University.

## Sex as a biological variable

Our study examined male and female animals, and similar findings are reported for both sexes.

## Paw plantar-incision model of post-surgical pain

Paw plantar-incision was performed as described in previous studies (*Pogatzki and Raja, 2003*). A 5-mm longitudinal incision was made through the skin and fascia of the plantar aspect, beginning 2 mm from the proximal edge of the heel and extending toward the toes. The flexor muscle was elevated with curved forceps.

## Animal behavioral tests

### Hargreaves test for heat hyperalgesia

The paw withdrawal latency was tested as described in our previous study (*Liu et al., 2019*) with 30 s as the cutoff time. Each hind paw was stimulated three times at an interval >5 min.

## Randall–Selitto test for mechanical hyperalgesia

The test consisted of applying the increasing mechanical force using the tip of Randall Selitto (IITC 2500) apparatus to the dorsal surface of the mouse hind paw. A total of three repeated tests were performed for each paw. Animal responses, including discomfort/struggle, paw withdrawal, and vocal responses, were observed as an endpoint of the result. The force resulting in any of the end-point behaviors was considered as the mechanical threshold. The cutoff force was $250 \times g$.

## CatWalk gait analysis

CatWalk XT version 10.6 gait analysis system (Noldus Information Technology, Wageningen, Netherlands) was used (*Feehan and Zadina, 2019*). At least three compliant runs were collected at each time point. The following parameters were investigated: (1) Print area, (2) Max contact area, and (c) Max contact intensity. To rule out the confounding influences of body weight and paw size, the walking parameters of the LH (injured side) were normalized by that of RH (uninjured side).

## Open field test

Locomotor activity was monitored and quantified using a Photobeam Activity System-Open Field (PAS-OF) (San Diego Instruments, San Diego, CA). The total number of beam breaks over the second 30 min was analyzed. On post-injury Day 2, the first round of open field test was conducted without drug treatment. On post-injury Day 4, mice received an intra-paw injection of the vehicle or FLO 30 min prior to the test.

## In vivo calcium imaging in mice

The L4 DRG of $Pirt^{Cre};Rosa26^{lsl-GCaMP6s}$ mice were exposed for imaging as described in our previous studies (*Gao et al., 2021*; *Zheng et al., 2022*). Mice under anesthesia (1.5% isoflurane) were laid on a custom-designed microscope stage with the spinal column being stabilized. Live images of the intact DRG were acquired at five frames with 600 Hz in frame-scan mode, using a 10×/0.30 long-working distance air objective lens (Leica, 506505) of a confocal microscope (Leica TCS SP8, Wetzlar, Germany). Dipping the hind paw into a 51°C water bath was applied as noxious heat stimulation.

Raw image stacks were imported into FIJI (NIH) for imaging data analysis. To measure the maximum fluorescence intensity ($F_t$), the average pixel values in a given region of interest were calculated for each image frame recorded during the whole recording period. A ratio of fluorescence difference ($\Delta F = F_t - F_0$) to baseline level ($F_0$) ≥30% was defined as an activation of the neuron. '% of total' represented the proportion of activated neurons relative to the total number of neurons counted from the same analyzed image. Somal areas of <450, 450–700, and >700 μm$^2$ were used for defining small, medium, and large DRG neurons, respectively (*Chen et al., 2022a*; *Chen et al., 2022b*).

## Immunocytochemistry

Mice were deeply anesthetized and perfused with 30 ml 0.1 M phosphate-buffered saline (PBS) (pH 7.4, 4°C) followed by 30 ml paraformaldehyde (PFA) solution 4% (vol/vol) in PBS (4°C) (*Green et al., 2019*). DRG and skin were dissected and post-fixed in 4% PFA at 4°C for 2 hr, then sectioned (15 μm width) with a cryostat. The slides were pre-incubated in a blocking solution and stained with indicated primary antibodies and corresponding secondary antibodies.

The slides were pre-incubated in blocking solution and stained against Griffonia simplicifolia isolectin GS-IB4 Alexa 568 (Invitrogen, I21412; 1:500), Neurofilament 200 (Sigma-Aldrich, N5389; 1:500), CGRP (Cell Signaling Technology, 14959; 1:200), Na$^+$/K$^+$ ATPase α-1 (Sigma-Aldrich, 05-369; 1:200), GFAP (Millipore, MAB360; 1:500), CD44 (Cell Signaling Technology, 3570; 1:500) and corresponding Alexa Fluor-conjugated secondary antibodies (1:500, Thermo Fisher Scientific). Raw confocal (TIFF) images (LSM 700; Zeiss, White Plains, NY, USA) were analyzed with Fiji (NIH). The total number of neurons in each section was determined by counting both labeled and unlabeled cell bodies. Positively stained neurons had clear stomata and an increase in fluorescence intensity ≥30% of the background. To quantify the neuronal cross-sectional area of DRG neurons, cells were identified by morphology with a clearly defined, dark, condensed nucleus. Positively stained cells were chosen for cross-sectional area measurement. The soma of the labeled cells was traced manually with the Fiji 'Freehand selection' tool, and the areas were measured. Tissues from different groups were processed together.

## Immunoblotting

The tissues were lysed in radioimmunoprecipitation assay buffer (Sigma, St. Louis, MO) containing a protease/phosphatase inhibitor cocktail (Cell Signaling Technology, Boston, MA). Samples (20 μg) were separated on a 4–12% Bis-Tris Plus gel (Thermo Fisher Scientific) and then transferred onto a polyvinylidene difluoride membrane (Thermo Fisher Scientific). Immunoreactivity was detected by enhanced chemiluminescence (Bio-Rad, Hercules, CA) after incubating the membranes with the indicated primary antibody (4°C, overnight).

Antibodies were chosen based on previous findings and our own study. GAPDH (EMD Millipore, 1:100,000) was used as an internal control for protein loading. CD44 (EMD Millipore, MABF580; 1:2000) was validated by a previous study (*Protin et al., 1999*). ImageJ (ImageJ 1.46r) was used to quantify the intensity of immunoreactive bands of interest on autoradiograms.

## In vitro calcium imaging

Experiments were conducted as described in our previous study (*Zhang et al., 2022*). Neurons were loaded with the fluorescent calcium indicator Fura-2-acetomethoxyl ester (2 μg/ml, Molecular Probes, Eugene, OR) for 45 min in the dark at room temperature and then allowed to de-esterify for 15 min at 37°C in the warm external solution. Cells were imaged at 340 and 380 nm excitation for the detection of intracellular free calcium.

## Cell viability assay (MTT assay)

Cell viability was evaluated by the MTT assay (Roche, 11465007001). DRG cells were seeded in 96-well microplates and subsequently exposed to several concentrations of HC-HA/PTX3 (0.5 μg to 15 μg/ml) with an incubation time of 24 hr at 37°C. After the treatments, the medium was removed and 10 μl of the MTT labeling reagent (final concentration 0.5 mg/ml) was added to each well. Optical density was measured in a spectrophotometer (Molecular Devices, FlexStation 3 Multi-Mode Microplate Reader) at 490 and 650 nm. Cell survival was expressed as the percentage of formazan absorbance, compared to the pretreatment level (experimental/control).

## Immunofluorescence of CD44 and F-actin in DRG neurons

Cultured lumbar DRGs were plated on an 8-mm glass coverslip. DRG neurons were then exposed to HC-HA/PTX3 (10 or 15 μg/ml with or without latrunculin-A [LAT-A, 1 μM, Invitrogen, L12370]) for 45 min at 37°C. Cells were fixed for 10 min in 4% PFA. The cover slips were then permeabilized with 0.3% Triton X-100 and sequentially stained with rat anti-CD44 antibody (BD Biosciences, 550538; 1:200), and corresponding Alexa 488-conjugated secondary antibodies (Invitrogen, A-11006; 1:500). The F-actin was stained with Alexa 568-conjugated phalloidin (Invitrogen, A12380; 1:400).

Raw confocal images (TIFF) were analyzed with Fiji (NIH). To quantify the changes in fluorescence of phalloidin and CD44 after drug treatment, the positive staining distributed along the whole plasma membrane and localized within the cell cytoplasm were traced manually with the Fiji 'Freehand selection' tool and the fluorescence intensity were measured. The proportion of staining on the plasma membrane was determined as a percent of the total staining measured in a cell.

## Electrophysiology

### Whole-cell patch-clamp recording of DRG neurons

Patch-clamp electrodes were conducted as described in our previous study (*Ford et al., 2022*). Briefly, for current-clamp recordings of intrinsic excitability, neurons were perfused with an oxygenated solution composed of (in mM) 140 NaCl, 4 KCl, 2 MgCl$_2$, 2 CaCl$_2$, 10 2-[4-(2-hydroxyethyl)piperazin-1-yl] ethanesulfonic acid (HEPES), and 10 glucose (pH = 7.4; ~305–310 mOsm). The internal solution was composed of (in mM) 135 K-Gluconate, 10 KCl, 10 HEPES, 2 Na$_2$ATP, 0.4 Na$_2$GTP, and 1 MgCl$_2$ (pH = 7.4 with KOH; ~300–305 mOsm).

For $I_{Nav}$ recordings, neurons were perfused with an oxygenated solution consisting of (in mM): 80 NaCl, 50 Choline-Cl, 30 TEA-Cl, 2 CaCl$_2$, 0.2 CdCl$_2$, 10 HEPES, 5 glucose (pH = 7.3; ~310–320 mOsm). The internal solution was composed of (in mM): 70 CsCl$_2$, 30 NaCl, 30 TEA-Cl, 10 EGTA, 1 CaCl$_2$, 2 MgCl$_2$, 2 Na$_2$ATP, 0.05 Na-GTP, 10 HEPES, 5 glucose (pH = 7.3 adjusted with CsOH; ~310 mOsm). For HVA-$I_{Ca}$ recordings, neurons were perfused with an oxygenated solution consisting of (in mM) 130 *N*-methyl-D-glucamine chloride (NMDG-Cl; solution of 130 mM NMDG, pH = 7.4), 5 BaCl$_2$, 1 MgCl$_2$,

10 HEPES, and 10 glucose (pH = 7.4; ~310–315 mOsm adjusted with sucrose). The internal solution was composed of (in mM) 140 TEA-Cl, 10 EGTA, 1 MgCl$_2$, 10 HEPES, 0.5 Na$_2$GTP, and 3 Na$_2$ATP (pH = 7.4; ~300–305 mOsm). In some experiments, 0.5 nM LAT-A or vehicle was added to the internal solution and infused into the neuron via the patch pipette.

All recordings were filtered at 4 kHz, sampled at a rate of 20 kHz, and stored on a personal computer (Dell) using pClamp 11 and a digitizer (Digidata 1550B, Molecular Devices). Currents were digitally filtered offline by using a low-pass Gaussian filter with a −3 dB cut-off set to 2 kHz (Clampfit software; pClamp 11, Molecular Devices).

## Intrinsic excitability studies of DRG neurons

After obtaining whole-cell configuration in both small (≤20 μm diameter) and large DRG neurons (≥30 μm diameter) (**Ford et al., 2022**) from mice after bilateral plantar-incision, a 5-min equilibration period was allowed. First, the spontaneous activity of the neuron was recorded for 1–2 min from $V_{rest}$. Additional intrinsic excitability measurements were then made before and after the bath application of 10 μg/ml HC-HA/PTX3. Rheobase was measured by injecting a series of square-wave current steps via the patch electrode (500 ms, 10 pA steps) until a single AP was generated. Additionally, pre-drug AP threshold (mV), AP amplitude (mV), AP half-width (ms), and input resistance (MΩ) were measured before and after drug application. All measurements were compared using paired $t$-tests.

## $I_{Nav}$ studies of DRG neurons

Under voltage-clamp conditions, whole-cell $I_{Nav}$ currents were normalized to each cell capacitance measurement to examine current density (pA/pF). For examination of $I_{Nav}$ current–voltage (I–V) relationships and steady-state activation, after 5-min infusion of 0.5 nM LAT-A or vehicle, a series of 50 ms depolarizing square wave voltage steps were delivered via the patch electrode ($V_{hold}$ = −90 mV; Vt$_{test}$ = −80 to +60 mV, 10 mV steps). 10 μg/ml HHP was then applied via bath perfusion for 4 min before the $I_{Nav}$ I–V protocol was run again. The current–voltage relationship (I–V curve) was ascertained by plotting normalized peak $I_{Nav}$ amplitudes at each test voltage (−80 to +60 mV). The voltage dependency of $I_{Nav}$ steady-state activation was determined by plotting normalized peak Na conductances ($G_{Na}/G_{max}$) at each test voltage. $G_{Na}$ was computed by: $G_{Na} = I_{Nav}/(V − V_{rev})$, where $I_{Nav}$ is the maximum sodium current during test voltage application. Data were then normalized to the maximum Na conductance ($G_{max}$), then fitted to a Boltzmann distribution:

$$G_{Na}/G_{max} = \frac{1}{1 + \exp\left[ze\dfrac{(V − V_{half})}{kT}\right]}$$

where $V_{half}$ is the potential for half max activation, $k$ is the Boltzmann constant, $z$ is an apparent gating charge, $T$ is the absolute temp, and $kT/e$ = 25 mV at 22°C. $I_{Nav}$ I–V relationships and normalized conductances were compared using two-way RM analysis of variance (ANOVA).

## HVA-I$_{Ca}$ studies of DRG neurons

In small-diameter DRG neurons (≤20 μm), whole-cell HVA-I$_{Ca}$ currents were normalized to each cell capacitance measurement to examine current density (pA/pF). For examination of HVA-I$_{Ca}$ current–voltage (I–V) relationships and channel open probabilities, after a 5-min infusion of 0.5 nM LAT-A or vehicle, a series of 25 ms depolarizing square wave voltage steps were delivered via the patch electrode ($V_{hold}$ = −80 mV; $V_{test}$ = −70 to +40 mV, 10 mV step). 10 μg/ml HC-HA/PTX3 was then applied via bath perfusion for 4 min before the HVA-I$_{Ca}$ I–V protocol was run again. The current–voltage relationships (I–V curves) were ascertained by plotting normalized peak HVA-I$_{Ca}$ amplitudes at each test voltage (−70 to +40 mV). The voltage dependency of HVA-I$_{Ca}$ channel open probability was determined by plotting normalized tail currents as a function of test voltages applied, which were then fitted with a Boltzmann equation for channel open probabilities:

$$P(V) = P_{min} + \frac{P_{max} − P_{min}}{1 + e^{\dfrac{V − V_{half}}{k}}}$$

$P(V)$ represents the channel open probability as a function of membrane potential; $P_{min}$ and $P_{max}$ are the minimum and maximum open probabilities; $V_{half}$ is the voltage at 50% maximum current; and $k$ is the default slope value.

## Nucleofection

The dissociated neurons from lumbar DRGs were suspended in 100 µl of Amaxa electroporation buffer (Lonza Cologne GmbH, Cologne, Germany) with siRNAs (0.2 nmol per transfection). Suspended cells were transferred to a 2.0-mm cuvette and electroporated with the Amaxa Nucleofector apparatus. After electroporation, cells were immediately mixed to the desired volume of prewarmed culture medium and plated on precoated coverslips or culture dishes.

## Quantitative PCR

Total RNA was isolated using PicoPure RNA Isolation Kit (Thermo Fisher Scientific) following the manufacturer's manual. RNA quality was verified using the Agilent Fragment Analyzer (Agilent Technologies, Santa Clara, CA). Two-hundred ng of total RNA was used to generate the cDNA using the SuperScript VILO MasterMix (Invitrogen, Waltham, MA). 10 ng of cDNA was run in a 20-µl reaction volume (triplicate) using PowerUp SYBR Green Master Mix to measure real-time SYBR green fluorescence with QuantStudio 3 Real-Time PCR Systems (Thermo Fisher Scientific). Calibrations and normalizations were performed using the $2^{-\Delta\Delta CT}$ method. Mouse *Gapdh* was used as the reference gene.

## FLO

FLO (Clarix Flo; BioTissue, Miami, FL) is a sterile, micronized, and lyophilized form of human AM and UC matrix used for surgical and non-surgical repair, reconstruction, or replacement of soft tissue by filling in the connective tissue void. Clarix Flo is regulated under section 361 of the Public Health Service Act and the regulations in 21 CFR Part 1271. Clarix Flo is derived from donated human placentas delivered from healthy mothers and is then aseptically processed to devitalize all living cells but retain the natural characteristics of the tissue.

## Qualification and release of HC-HA/PTX3

HC-HA/PTX3 was purified from human AM after donor eligibility was determined according to the requirements by FDA based on our published method (*He et al., 2009*) with modifications and was performed according to good laboratory practices (GLP). All standard operating procedures, work instructions, and forms used for release testing were approved by the quality assurance department of Biotissue, inc and the testing was performed according to GLP. The purity of HC-HA/PTX3 was disclosed by the lack of detectable proteins per BCA assay with a detectable level of 11.7 ± 3.2 µg/ml and notable reduction of protein bands per silver staining in AM4P when compared to AM2P with or without NaOH, which cleaves the ester bond between HA and HC1 (*Figure 3—figure supplement 1A*). Due to the lack of detectable proteins, HC-HA/PTX3 was tested based on the amount of HA. HA in HC-HA/PTX3 was of high molecular weight (HMW) (≥500 kDa) when compared to the HMW-HA control using agarose gel electrophoresis (*Figure 3—figure supplement 1B*). It was released by confirming the identity of HC-HA/PTX3 based on Western blot analysis using respective antibody specific to HC1 (ITIH1 antibody, Cat# ab70048, Abcam, Waltham, MA, USA) and PTX3 (PTX3 antibody, Cat# ALX-804-464-C100, Enzo, Farmingdale, NY, USA) with or without hyaluronidase (HAase) digestion to release HC1 and HMW PTX3 from HC-HA/PTX3 in the loading well into the gel and with or without reduction by DTT, which further rendered PTX3 from HMW (octamer) to dimer and monomer (*Figure 3—figure supplement 1C, D*). In addition, each batch of HC-HA/PTX3 was released after it also passed the potency assay (*Figure 3—figure supplement 1E*), with the acceptance criterion of no less than 89.21% inhibition of tartrate-resistant acid phosphatase (TRAP) activity of osteoclast differentiation in cloned monocytes of murine RAW264.7 cell line (ATCC, Manassas, VA, USA) by receptor activator of nuclear factor kappa-Β ligand (RANKL) (PeproTech, Cranbury, NJ, USA).

## Culturing DRG neurons

Lumbar DRGs from 4-week-old WT mice and CD44 KO mice (both sexes) were collected in cold DH10 (*Weng et al., 2015*) (90% Dulbecco's Modified Eagle Medium [DMEM]/F-12, 10% fetal bovine serum, penicillin [100 U/ml], and streptomycin [100 µg/ml] [Invitrogen]) and treated with enzyme solution

(dispase [5 mg/ml] and collagenase type I [1 mg/ml] in Hanks' balanced salt solution without $Ca^{2+}$ or $Mg^{2+}$ [Invitrogen]) for 35 min at 37°C. After trituration, the supernatant with cells was filtered through a Falcon 40 µm (or 70 µm) cell strainer. Then the cells were spun down with centrifugation and were resuspended in DH10 with growth factors (25 ng/ml nerve growth factor [NGF]; 50 ng/ml glial cell line-derived neurotrophic factor [GDNF]), plated on glass coverslips coated with poly-D-lysine (0.5 mg/ml; Biomedical Technologies Inc) and laminin (10 µg/ml; Invitrogen), cultured in an incubator (95% $O_2$ and 5% $CO_2$) at 37°C.

## Statistical analyses

Statistical analyses were performed with the Prism 9.0 statistical program (GraphPad Software, Inc). The methods for statistical comparisons in each study were indicated in the figure legends. To reduce selection and observation bias, animals were randomized to the different groups and the experimenters were blinded to drug treatment. The comparisons of data consisting of two groups were made by Student's *t*-test. Comparisons of data in three or more groups were made by one-way ANOVA followed by the Bonferroni post hoc test. Comparisons of two or more factors across multiple groups were made by two-way ANOVA followed by the Bonferroni post hoc test. Two-tailed tests were performed, and $p < 0.05$ was considered statistically significant in all tests.

## Study approval

This study was performed in strict accordance with the recommendations in the Guide for the Care and Use of Laboratory Animals of the National Institutes of Health. All of the animals were handled and studies were conducted according to approved protocols by the Johns Hopkins University Animal Care and Use Committee (Baltimore, MD, USA) to ensure minimal animal use and discomfort (Protocol: MO22M306).

## Acknowledgements

The authors thank Kristy Hamlin, Monica Cuellar Entrena, and Melissa Suarez from BioTissue, Inc for their work preparing and characterizing HC-HA/PTX3. The authors also thank Drs. Courtney McQueen and Anne N Connor (Senior Science Writers, Research Development Team, Office of the Vice Provost for Research, Johns Hopkins University) for editing the manuscript. This study was supported by the National Institutes of Health (NIH, Bethesda, Maryland, USA) grants NS110598 (YG) and NS117761 (YG), and by the Lerner Family Fund for Pain Research (NCF). X-WW was supported by an NIH grant K99EY031742. Funders had no role in study design, data collection, data interpretation, or the decision to submit the work for publication. YG and SNR received research grant support from Medtronic, Inc.

## Additional information

### Competing interests

Megha Mahabole, Hua He, Scheffer C Tseng: employed by BioTissue, Inc. Dazhi Yang: employed by Acrogenic Technologies Inc. The other authors declare that no competing interests exist.

### Funding

| Funder | Grant reference number | Author |
| --- | --- | --- |
| National Institute of Neurological Disorders and Stroke | NS110598 | Yun Guan |
| National Institute of Neurological Disorders and Stroke | NS117761 | Yun Guan |
| National Eye Institute | K99EY031742 | Xuewei Wang |

| Funder | Grant reference number | Author |
|--------|------------------------|--------|

The funders had no role in study design, data collection, and interpretation, or the decision to submit the work for publication.

## Author contributions

Chi Zhang, Qian Huang, Neil C Ford, Data curation, Formal analysis, Methodology, Writing - original draft, Writing – review and editing; Nathachit Limjunyawong, Resources, Data curation, Methodology; Qing Lin, Conceptualization, Data curation, Methodology; Fei Yang, Xiang Cui, Ankit Uniyal, Jing Liu, Xuewei Wang, Irina Duff, Yiru Wang, Jieru Wan, Data curation, Formal analysis, Methodology; Megha Mahabole, Hua He, Resources, Methodology; Guangwu Zhu, Resources, Methodology, Project administration; Srinivasa N Raja, Conceptualization, Supervision, Investigation, Writing – review and editing; Hongpeng Jia, Conceptualization, Supervision, Methodology, Writing – review and editing; Dazhi Yang, Resources, Methodology, Writing – review and editing; Xinzhong Dong, Supervision, Methodology, Writing – review and editing; Xu Cao, Conceptualization, Supervision, Writing – review and editing; Scheffer C Tseng, Conceptualization, Resources, Methodology, Writing – review and editing; Shaoqiu He, Data curation, Formal analysis, Methodology, Writing – review and editing; Yun Guan, Conceptualization, Resources, Supervision, Funding acquisition, Investigation, Methodology, Writing - original draft, Project administration, Writing – review and editing

## Author ORCIDs

Xinzhong Dong  https://orcid.org/0000-0002-9750-7718
Xu Cao  https://orcid.org/0000-0001-8614-6059
Yun Guan  https://orcid.org/0000-0003-1321-6655

## Ethics

This study was performed in strict accordance with the recommendations in the Guide for the Care and Use of Laboratory Animals of the National Institutes of Health. All of the animals were handled and studies were conducted according to approved protocols by the Johns Hopkins University Animal Care and Use Committee (Baltimore, MD, USA) to ensure minimal animal use and discomfort (Protocol: MO22M306).

Reviewer #1 (Public review): https://doi.org/10.7554/eLife.101269.3.sa1
Reviewer #2 (Public review): https://doi.org/10.7554/eLife.101269.3.sa2
Reviewer #3 (Public review): https://doi.org/10.7554/eLife.101269.3.sa3
Author response https://doi.org/10.7554/eLife.101269.3.sa4

# Additional files

## Supplementary files

• Supplementary file 1. The measures of intrinsic membrane properties of small-diameter dorsal root ganglion (DRG) neurons in wild-type (WT) and CD44 knockout (KO mice). Knocking out of CD44 did not significantly alter the intrinsic membrane property of DRG neurons, as compared to that in WT mice.

• MDAR checklist

## Data availability

All data generated or analyzed during this study are included in the manuscript, figures, figure supplements, and source data files.

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
