## [Editor Report · eLife Assessment]

The authors provide **convincing** data that identify a novel, non-opioid biologic from human birth tissue products with anti-nociceptive properties in a preclinical mouse model of surgical pain. This **important** study highlights the potential use of naturally derived biologics from human birth tissues as safe and sustainable pain treatment options that do not possess the adverse side effects associated with opioids and synthetic pharmaceuticals. Whether these results will translate to the clinic remains to be seen, nevertheless, these preclinical findings are promising.

---

## [Referee Report · Reviewer #1 (Public review)]

Summary:

Opioids and related drugs are powerful analgesics that reduce suffering from pain. Unfortunately, their use often leads to addiction and there is an opioid-abuse epidemic that affects people worldwide. This study represents an ongoing effort to develop non-opioid analgesics for pain management. The findings point to an alternative approach to control post-surgical pain in lieu of opioid medications.

Strengths:

(1) The study responds to the urgent need for the development of non-opioid analgesics.

(2) The study demonstrates the efficacy of Clarix Flo (FLO) and HC-HA/PTX3 from the human amniotic membrane (AM) in reducing pain in a mouse model without the adverse effects of opioids.

(3) The study further explored the underlying mechanisms of how HC-HA/PTX3 produces its effects on neurons, suggesting the molecules/pathways involved in pain relief.

(4) The potential use of naturally derived biologics from human birth tissues (AM) is safe and sustainable, compared to synthetic pharmaceuticals.

(5) The study was conducted with scientific rigor, involving purification of active components, comparative analysis with multiple controls, and mechanistic explorations.

Weaknesses:

(1) It should be cautioned that while the preclinical findings are promising, these results still need to be translated into clinical settings that are complex and often unpredictable.

(2) The study shows the efficacy of FLO and HC-HA/PTX3 in one preclinical model of post-surgical pain. The observed effect may be variable in other pain conditions.

Comments on revisions:

The authors have addressed my concerns in the revision. I don't have further comments on this manuscript.

---

## [Referee Report · Reviewer #2 (Public review)]

Summary:

This is an outstanding piece of work on the potential of FLO as a viable analgesic biologic for the treatment of postsurgical pain. The authors purified the HC-HA/PTX3 from FLO and demonstrated its potential as an effective non-opioid therapy for postsurgical pain. They further unraveled the mechanisms of action of the compound at cellular and molecular levels.

Strengths:

Prominent strengths include the incorporation of behavioral assessment, electrophysiological and imaging recordings, the use of knockout and knockdown animals, and the use of antagonist agents to verify biological effects. The integrated use of these techniques, combined with the hypothesis-driven approach and logical reasoning, provides compelling evidence and novel insight into the mechanisms of the significant findings of this work.

Weaknesses:

I did not find any significant weaknesses even with a critical set of mind. The only minor suggestion is that the Results section may focus on the results from this study and minimize the discussions of background information.

Comments on revisions:

The authors have adequately addressed all the points raised in the last round of review. Thanks!

---

## [Referee Report · Reviewer #3 (Public review)]

Summary:

Non-opioid analgesics derived from human amniotic membrane (AM) product represents a novel and unique approach to analgesia that may avoid the traditional harms associated with opioids. Here, the study investigators demonstrate that HC-HAPTX3 is the primary bioactive component of the AM product FLO responsible for anti-nociception in mouse-model and in-vitro dorsal root ganglion (DRG) cell culture experiments. The mechanism is demonstrated to be via CD44 with an acute cytoskeleton rearrangement that is induced that inhibits Na+ and Ca++ current through ion channels. Taken together, the studies reported in the manuscript provide supportive evidence clarifying the mechanisms and efficacy of HC-HAPTX3 antinociception and analgesia.

Strengths:

Extensive experiments including murine behavioral paw withdrawal latency and Catwalk test data demonstrating analgesic properties. Breadth and depth of experimental data are clearly supporting mechanisms and antinociceptive properties.

Weaknesses:

None. Only a few minor directed changes to the text of the manuscript.

P4 last sentence - "Our findings highlight the potential of a naturally derived biologic from human birth tissues as an effective non-opioid treatment for post-surgical pain and unravel the underlying mechanisms." - another sentence clause is required before "unravel"

P7 second paragraph - please edit the following sentence for clarity: "Since HC-HA/PTX3 mimics FLO in producing pain inhibition, and it has high-purity and is more water-soluble than FLO, making it suitable for probing cellular mechanisms."

---

## [Author Response]

The following is the authors’ response to the original reviews.

**Public Reviews:**

**Reviewer #1 (Public review):**
Summary:Opioids and related drugs are powerful analgesics that reduce suffering from pain. Unfortunately, their use often leads to addiction and there is an opioid-abuse epidemic that affects people worldwide. This study represents an ongoing effort to develop non-opioid analgesics for pain management. The findings point to an alternative approach to control post-surgical pain in lieu of opioid medications.Strengths:(1) The study responds to the urgent need for the development of non-opioid analgesics.(2) The study demonstrates the efficacy of Clarix Flo (FLO) and HC-HA/PTX3 from the human amniotic membrane (AM) in reducing pain in a mouse model without the adverse effects of opioids.(3) The study further explored the underlying mechanisms of how HC-HA/PTX3 produces its effects on neurons, suggesting the molecules/pathways involved in pain relief.(4) The potential use of naturally derived biologics from human birth tissues (AM) is safe and sustainable, compared to synthetic pharmaceuticals.(5) The study was conducted with scientific rigor, involving purification of active components, comparative analysis with multiple controls, and mechanistic explorations.Weaknesses:(1) It should be cautioned that while the preclinical findings are promising, these results still need to be translated into clinical settings that are complex and often unpredictable.(2) The study shows the efficacy of FLO and HC-HA/PTX3 in one preclinical model of post-surgical pain. The observed effect may be variable in other pain conditions.

We thank the reviewer for these good comments and support! We agree with your suggestions and have provided more information in the discussion (Pages 11-12) and conclusion to address these comments.

**Reviewer #2 (Public review):**
Summary:This is an outstanding piece of work on the potential of FLO as a viable analgesic biologic for the treatment of postsurgical pain. The authors purified the HC-HA/PTX3 from FLO and demonstrated its potential as an effective non-opioid therapy for postsurgical pain. They further unraveled the mechanisms of action of the compound at cellular and molecular levels.Strengths:Prominent strengths include the incorporation of behavioral assessment, electrophysiological and imaging recordings, the use of knockout and knockdown animals, and the use of antagonist agents to verify biological effects. The integrated use of these techniques, combined with the hypothesis-driven approach and logical reasoning, provides compelling evidence and novel insight into the mechanisms of the significant findings of this work.Weaknesses:I did not find any significant weaknesses even with a critical mindset. The only minor suggestion is that the Results section may focus on the results from this study and minimize the discussions of background information.

We thank the reviewer for your support! We revised the result section as suggested and reduced the discussion of background information.

**Reviewer #3 (Public review):**
Summary:Non-opioid analgesics derived from human amniotic membrane (AM) product represents a novel and unique approach to analgesia that may avoid the traditional harms associated with opioids. Here, the study investigators demonstrate that HC-HAPTX3 is the primary bioactive component of the AM product FLO responsible for anti-nociception in mouse-model and in-vitro dorsal root ganglion (DRG) cell culture experiments. The mechanism is demonstrated to be via CD44 with an acute cytoskeleton rearrangement that is induced that inhibits Na+ and Ca++ current through ion channels. Taken together, the studies reported in the manuscript provide supportive evidence clarifying the mechanisms and efficacy of HC-HAPTX3 antinociception and analgesia.Strengths:Extensive experiments including murine behavioral paw withdrawal latency and Catwalk test data demonstrating analgesic properties. The breadth and depth of experimental data are clearly supporting mechanisms and antinociceptive properties.Weaknesses:A few changes to the text of the manuscript would be recommended but no major weaknesses were identified.

We thank the reviewer for your support! We revised these texts as suggested.

**Recommendations for the authors:Reviewer #1 (Recommendations for the authors):**
(1) The study showed an effect on baseline nociception and acute post-surgical pain. Chronic post-surgical pain is a major problem and should be considered.

We thank the reviewer for this comment. To further improve the translational potential, we will extend current findings and employ chronic post-surgical pain models, such as skin/muscle incision and retraction (SMIR) in the thigh of the rodent,(1-3) as well as chronic pain models such as neuropathic pain in the future. We acknowledged this limitation in the discussion. (Page 12)

(2) Indicate the source of cultures DRGs.

We added “Method 15 Culturing DRG neurons” in the revised manuscript.

(3) The size of DRG neurons was described in cross-sectional area (Figure 2 caption) and diameter (method). Be consistent.

We thank the reviewer for this comment. Cross-sectional area has often been used for describing the size of DRG neurons for in vivo calcium imaging studies, including our previous work (4, 5). In order to keep consistent and make data comparable between studies, we also used the cross-sectional area in current study in Fig 2 in vivo calcium imaging experiment. On the other hand, cell-diameter has been routinely/widely used for in vitro experiments such as in vitro electrophysiology recording and immunofluorescence staining of cultured DRG neurons. To be consistent with this tradition, we used cell-diameter in these experiments. Methods for measuring the area and diameter are explicitly described for each experimental setting, and consistent between the current study and our previous studies (6). In the manuscript, our previously published studies have also been cited in the Methods section. (Method “4 In vivo calcium imaging in mice” and “10.2 Intrinsic excitability studies of DRG neurons”).

(4) Clarify what "% of total" means in Figure 2. For bar graphs in 2B-D, the percent of total activated neurons (small, medium, and large) does not add up to 100.

“% of total” represented the proportion of activated neurons relative to the total number of neurons counted from the same analyzed image. This information was added to the figure legend of Figure 2 (B-C) and Method “4 In vivo calcium imaging in mice” in the revised manuscript. At the end of each experiment, we can over-exposure the image to unravel all neuronal profiles and count the total number of neurons on that field/image. Only a small portion of neurons in each size category responded to the test stimulation, and hence the total does not add up to 100.

(5) Discuss clinical data or human studies to validate the efficacy and safety of FLO or HC-HA/PTX3 in patients.

Thanks for the great suggestion. We provided a brief discussion (Page 11-12).

Cryopreserved AM/UC has been clinically validated through several hundred peer-reviewed publications since 1995, including 12 studies specifically assessing FLO (Clarix Flo). These studies collectively support the safety and preliminary effectiveness of Clarix Flo in managing some clinical pain conditions such as knee osteoarthritis(7, 8), discogenic pain (9), rotator cuff tears(10), and painful neuropathy of the lower extremities (11). Currently, HC-HA/PTX3 is limited to pre-clinical research, and to our knowledge, there are no available data on its clinical efficacy and safety.

(6) Introduction, last sentence of the second paragraph, delete "also".

Thanks for carefully examining our manuscript. It was revised as suggested.

**Reviewer #2 (Recommendations for the authors):**
My only recommendation for improving the writing and presentation is to shorten the discussion of background information in Results.

We thank the reviewer for your support and comments! We previously intended to provide some background information to help readers understand the premise and rationale of the study, before illustrating our findings. Nevertheless, we reduced some background information in the result section as suggested by this reviewer to make it more straightforward.

**Reviewer #3 (Recommendations for the authors):**
P4 last sentence - "Our findings highlight the potential of a naturally derived biologic from human birth tissues as an effective non-opioid treatment for post-surgical pain and unravel the underlying mechanisms." - another sentence clause is required before "unravel".

As advised, we revised the sentence to: “Collectively, our findings highlight the potential of naturally derived biologics from human birth tissues as an effective non-opioid treatment for post-surgical pain. Moreover, we unravel the underlying mechanisms of pain inhibition induced by FLO and HC-HA/PTX3.”

P7 second paragraph - please edit the following sentence for clarity: "Since HC-HA/PTX3 mimics FLO in producing pain inhibition, and it has high purity and is more water-soluble than FLO, making it suitable for probing cellular mechanisms.".

As advised, we have revised the sentence. “Since HC-HA/PTX3 mimics FLO in its ability to inhibit pain and has higher purity and greater water solubility compared to FLO, it is well-suited for investigating cellular mechanisms.”

References:

(1) Flatters SJ. Characterization of a model of persistent postoperative pain evoked by skin/muscle incision and retraction (SMIR). *Pain.* 2008;135(1-2):119-30.

(2) Ying YL, Wei XH, Xu XB, She SZ, Zhou LJ, Lv J, et al. Over-expression of P2X7 receptors in spinal glial cells contributes to the development of chronic postsurgical pain induced by skin/muscle incision and retraction (SMIR) in rats. *Experimental neurology.* 2014;261:836-43.

(3) Cao S, Bian Z, Zhu X, and Shen SR. Effect of Epac1 on pERK and VEGF Activation in Postoperative Persistent Pain in Rats. *Journal of molecular neuroscience : MN.* 2016;59(4):554-64.

(4) Chen Z, Huang Q, Song X, Ford NC, Zhang C, Xu Q, et al. Purinergic signaling between neurons and satellite glial cells of mouse dorsal root ganglia modulates neuronal excitability in vivo. *Pain.* 2022;163(8):1636-47.

(5) Chen Z, Zhang C, Song X, Cui X, Liu J, Ford NC, et al. BzATP Activates Satellite Glial Cells and Increases the Excitability of Dorsal Root Ganglia Neurons In Vivo. *Cells.* 2022;11(15).

(6) Ford NC, Barpujari A, He SQ, Huang Q, Zhang C, Dong X, et al. Role of primary sensory neurone cannabinoid type-1 receptors in pain and the analgesic effects of the peripherally acting agonist CB-13 in mice. *Br J Anaesth.* 2022;128(1):159-73.

(7) Castellanos R, and Tighe S. Injectable Amniotic Membrane/Umbilical Cord Particulate for Knee Osteoarthritis: A Prospective, Single-Center Pilot Study. *Pain Med.* 2019;20(11):2283-91.

(8) Mead OG, and Mead LP. Intra-Articular Injection of Amniotic Membrane and Umbilical Cord Particulate for the Management of Moderate to Severe Knee Osteoarthritis. *Orthop Res Rev.* 2020;12:161-70.

(9) Buck D. Amniotic Umbilical Cord Particulate for Discogenic Pain. *J Am Osteopath Assoc.* 2019;119(12):814-9.

(10) Ackley JF, Kolosky M, Gurin D, Hampton R, Masin R, and Krahe D. Cryopreserved amniotic membrane and umbilical cord particulate matrix for partial rotator cuff tears: A case series. *Medicine (Baltimore).* 2019;98(30):e16569.

(11) Buksh AB. Ultrasound-guided injections of amniotic membrane/umbilical cord particulate for painful neuropathy of the lower extremity. *Cogent Medicine.* 2020;7(1):1724067.